# Inference-Time Dynamic Modality Selection for Incomplete Multimodal Classification

**Siyi Du**[1*]    **Xinzhe Luo**[1]    **Declan P. O'Regan**[2]    **Chen Qin**[1*]

[1]Department of Electrical and Electronic Engineering & I-X, [2] MRC Laboratory of Medical Science
Imperial College London, London, UK

`{s.du23, x.luo, declan.oregan, c.qin15}@imperial.ac.uk`

## Abstract

Multimodal deep learning (MDL) has achieved remarkable success across various domains, yet its practical deployment is often hindered by incomplete multimodal data. Existing incomplete MDL methods either discard missing modalities, risking the loss of valuable task-relevant information, or recover them, potentially introducing irrelevant noise, leading to the ***discarding-imputation dilemma***. To address this dilemma, in this paper, we propose **DyMo**, a new inference-time dynamic modality selection framework that adaptively identifies and fuses reliable recovered modalities, fully exploring task-relevant information beyond the conventional discard-or-impute paradigm. Central to DyMo is a novel selection algorithm that maximizes multimodal task-relevant information for each test sample. Since direct estimation of such information at test time is intractable due to the unknown data distribution, we theoretically establish a connection between information and the task loss, which we compute at inference time as a tractable proxy. Building on this, a novel principled reward function is proposed to guide modality selection. In addition, we design a flexible multimodal network architecture compatible with arbitrary modality combinations, alongside a tailored training strategy for robust representation learning. Extensive experiments on diverse natural and medical image datasets show that DyMo significantly outperforms state-of-the-art incomplete/dynamic MDL methods across various missing-data scenarios. Our code is available at `https://github.com//siyi-wind/DyMo`.

## 1 Introduction

Multimodal / multi-view deep learning (MDL), which integrates various modalities/views to achieve a multisensory perception akin to humans, has gained increasing attention and made significant advances in various domains such as healthcare (Acosta et al., 2022), marketing (Liu et al., 2024), and embodied intelligence (Duan et al., 2022). However, real-world deployment of these MDL models remains limited due to their simplified data assumptions. Existing MDL approaches typically presuppose full modality availability during inference. In practice, however, samples often lack one or more modalities due to heterogeneous collection protocols across centers, sensor malfunctions, or transmission errors (Wu et al., 2024a), leading to degraded model performance. Therefore, developing MDL models robust to incomplete multimodal data has become a critical research focus.

Current methods addressing missing modality can be broadly categorized into two types: (1) *recovery-based* approaches aim to impute missing modalities at the input level or in latent space via retrieval or generation, enabling the MDL model to operate as if all modalities were present (Ma et al., 2021; Xu et al., 2025); (2) *recovery-free* approaches are designed to ignore missing modalities and make predictions using only available ones (Lee et al., 2023; Wu et al., 2024b).

However, these incomplete MDL methods encounter intrinsic challenges in capturing task-relevant information under modality heterogeneity (Zhang et al., 2024). Specifically, modalities vary in their task relevance, due to differences in the strength of task-relevant signals and the degree of interfering,

---

*Corresponding authors.

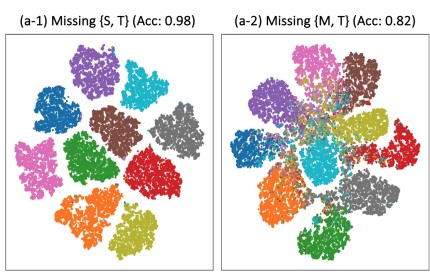
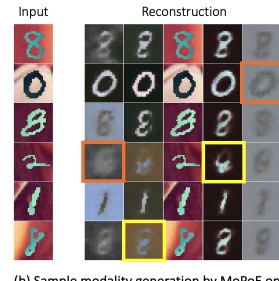
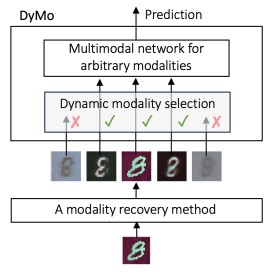

(a-1) Missing {S, T} (Acc: 0.98)  (a-2) Missing {M, T} (Acc: 0.82)

(a) t-SNE visualization of ModDrop's modality-incomplete features on the MNIST(M)-SVHN(S)-TEXT(T) (MST) training dataset

(b) Sample modality generation by MoPoE on the PolyMNIST test dataset

(c) Our DyMo: dynamic modality selection over recovered missing modalities at inference

Figure 1: (a-b) Evidence of the *discarding-imputation dilemma*: (a-1) *vs.* (a-2) recovery-free methods (*e.g.*, ModDrop (Neverova et al., 2015)) learn less discriminative features because they ignore highly task-relevant missing modalities {M,T}; (b) recovery-based methods (*e.g.*, MoPoE (Sutter et al., 2021)) generate unreliable reconstructions, *e.g.*, low-fidelity (orange) or misaligned (yellow). (c) Our DyMo, which addresses the dilemma by dynamically fusing task-relevant recovered modalities, improving accuracy by 1.61% on PolyMNIST, 1.68% on MST, and 3.88% on CelebA (Tab. 1).

task-irrelevant noise (Huang et al., 2022b). As shown in Fig. 1(a), when highly informative modalities are missing, earlier recovery-free methods rely solely on the less distinguishable features of the remaining modalities, without considering the valuable task-relevant information contained in the missing ones, resulting in decreased model performance. Prior recovery-based methods appear to mitigate this issue by reconstructing missing modalities. Yet, imputation qualities often vary across samples due to cross-modal heterogeneity and diverse missing scenarios. As in Fig. 1(b), some recovered modalities may be *low-fidelity* (*i.e.*, blurry or corrupted, orange boxes) or *semantically misaligned* (*i.e.*, their labels are inconsistent with the input modalities, yellow boxes). Integrating such unreliable modalities can inject task-irrelevant noise, impairing decision-making. Thus, discarding missing modalities risks losing valuable task-relevant information, whereas restoring them may inject harmful information, a trade-off limitation we term the ***discarding-imputation dilemma***.

To address this dilemma in incomplete MDL, we introduce a new perspective that dynamically selects and fuses recovered modalities conditioned on their task relevance, moving beyond the conventional dichotomy of discarding *vs.* imputing missing data. A key technical challenge in such dynamic systems is how to estimate the task-relevant informativeness of each modality and identify unreliable modalities at inference time. Nevertheless, existing dynamic fusion approaches (Cao et al., 2024; Gao et al., 2024) are primarily designed for low-fidelity modalities and rely on modality-specific features for weighting, and thus are limited in identifying semantically misaligned modalities.

In this work, we propose **DyMo**, a novel inference-time dynamic modality fusion framework that adaptively selects and fuses reliable recovered modalities to maximize multimodal task-relevant information in incomplete MDL (see Fig. 1(c)). To avoid integrating unreliable (low-fidelity or misaligned) recovery, we propose a novel dynamic algorithm that iteratively selects the most informative recovered modality based on the incremental multimodal task-relevant information gain it provides. Since the data distribution is unknown at inference, we theoretically derive that reducing task loss can increase the lower bound on task-relevant information. This motivates using the loss decrease as a practical proxy for information gain and introducing a novel principled reward function to guide modality selection. Additionally, to support flexible multimodal input, we design a multimodal network architecture capable of predicting task targets with arbitrary modality combinations. We further propose a tailored training strategy to learn robust latent features suitable for DyMo's dynamic selection process.

Our contributions can be summarized as follows. (1) To the best of our knowledge, we are the first to investigate the *discarding-imputation dilemma* in incomplete MDL and introduce dynamic neural networks to address it. (2) We propose DyMo, a novel dynamic framework that adaptively fuses recovered modalities via a new selection algorithm formulated on multimodal task-relevant information gain, together with a multimodal network and a tailored training algorithm for robust feature extraction from arbitrary combinations of modalities. (3) Experiments on 5 diverse datasets demonstrate DyMo's remarkable outperformance over incomplete/dynamic MDL SOTAs, especially under severe missing scenarios. DyMo is also easy-to-use (flexible with various modality recovery methods) and readily deployable without additional architecture overhead for its dynamic algorithm.

## 2    RELATED WORK

**Incomplete Multimodal Deep Learning (MDL)** methods (Zhan et al., 2025; Wu et al., 2024a) broadly fall into two categories: recovery-based and recovery-free. Recovery-based approaches can be further divided into offline and online, depending on when the imputation occurs. Offline methods typically pre-train separate reconstruction networks, *e.g.*, variational auto-encoders (VAEs) (Sutter et al., 2021; Gao & Pu, 2025), and then use reconstructed data for downstream tasks. Online methods (Wang et al., 2023b; Shou et al., 2025) jointly learn modality recovery and downstream tasks during training (Wang et al., 2023a). For instance, SMIL (Ma et al., 2022) employs Bayesian meta-learning to impute latent features, while M3Care (Zhang et al., 2022) retrieves similar samples to compensate for missing modalities. In recovery-free approaches, missing-agnostic techniques learn modality-invariant features via random modality dropout, *e.g.*, ModDrop (Neverova et al., 2015) and MMANet (Wei et al., 2023), or contrastive learning (Wu et al., 2024b), while missing-aware methods introduce missingness-specific parameters to handle different missing patterns (Ma et al., 2022; Li et al., 2025). However, previous incomplete MDL methods either fully recover or ignore missing modalities, without considering the *discarding-imputation dilemma*.

**Dynamic Deep Neural Networks**, in contrast to static models, can adapt their architectures and parameters based on individual input, yielding substantial improvements in computational efficiency, accuracy, and interpretability (Han et al., 2021; Guo et al., 2024). Dynamic architecture methods adjust network depth, width, or routing paths on a per-sample basis (Fedus et al., 2022; Yue et al., 2024; Bayasi et al., 2022; Zhao et al., 2025), while dynamic parameter algorithms modulate network weights or feature scaling without altering the architecture (Zhang et al., 2023; Bolya et al., 2023; Du et al., 2023). Our DyMo, which dynamically routes modality data conditioned on each instance, falls into the dynamic architecture category.

**Multimodal Fusion** is a critical research problem in MDL (Baltrušaitis et al., 2018). Existing methods can be grouped by their fusion stage into early (data level), intermediate (feature level), and late (decision level) fusion. Recently, dynamic multimodal fusion methods have emerged to address instance-specific modality reliability variations. These methods assign different weights to each modality (Zhang et al., 2023; Gao et al., 2024) or selectively use a subset of modalities (Xue & Marculescu, 2023; Ma et al., 2025). However, such methods typically assume all modalities are available and mainly focus on intra-modality noise (*i.e.*, low-fidelity data) during training, lacking robustness against inter-modality errors (*i.e.*, semantic misalignment). While MICINet (Zhang et al., 2025) explores both types of noise, it requires ground-truth labels and cannot be applied during inference. By contrast, DyMo dynamically fuses observed and recovered modalities to handle incomplete data, while effectively addressing intra- and inter-modality noise without relying on labels.

## 3    METHOD

In this section, we present DyMo, a novel inference-time dynamic modality selection framework for incomplete multimodal classification by fully exploring task-relevant information from recovered modalities, while addressing the *discarding-imputation dilemma* (Fig. 1(c)). To achieve this, DyMo comprises 3 key components: (1) a flexible multimodal architecture capable of making predictions from arbitrary modalities (Sec. 3.1, Fig. 2); (2) a novel dynamic selection algorithm that integrates valuable recovered modalities to maximize multimodal task-relevant information for each sample (Sec. 3.2, Algorithm 1); and (3) a tailored training strategy that enhances representation learning to ensure feature robustness under dynamic modality configuration during inference (Sec. 3.3).

### 3.1    MULTIMODAL ARCHITECTURE FOR ARBITRARY MODALITIES

Our multimodal network is designed to produce reliable predictions from any subsets of input modalities, enabling adaptive fusion based on the task relevance of each recovered modality. To achieve this, we construct a multimodal network $f$ (Fig. 2), consisting of modality-specific encoders for unimodal feature extraction, a multimodal transformer for modeling cross-modal interactions and learning a multimodal representation of available modalities, and a classifier for the final prediction.

A multimodal classification dataset can be defined as $\{\mathbb{X}_i, y_i\}_{i=1}^{N}$, where $N$ is the number of samples, $y_i \in \{1, ..., K\}$ is the class label, and $\mathbb{X}_i = \{\boldsymbol{x}^{(m)}\}_{m \in \mathcal{I}_i}$ represents a (potentially incomplete)

multimodal input comprising modalities such as images and text. Here, $\mathcal{I}_i$ denotes a subset of the complete modality indices $[M] = \{1, 2, ..., M\}$. As shown in Fig. 2, each available modality $m \in \mathcal{I}_i$ is encoded by a modality-specific encoder $h^{(m)}$, producing a sequence of feature tokens $\boldsymbol{H}^{(m)} \in \mathbb{R}^{L^{(m)} \times C}$, where $L^{(m)}$ is the sequence length and $C$ denotes the feature dimension. We then concatenate these tokens together with a learnable [CLS] token to form a multimodal sequence embedding. The [CLS] token's output embedding $\boldsymbol{z}$ serves as the multimodal representation for classification as in (Devlin et al., 2019). To preserve the sequence structure across samples, we assign dummy tokens to the positions of missing modalities.

Driven by transformer's ability to capture long-range dependencies from variable lengths of the sequence, we build a multimodal transformer network $\psi$ composed of $T$ stacked transformer layers (Vaswani et al., 2017) to learn cross-modal relations in observed modalities.

These transformer layers conduct self-attention on the multimodal sequence embedding and apply attention masks to ensure that missing modalities do not distort representation learning. The extracted multimodal representation $\boldsymbol{z} = \psi[h(\mathbb{X}_i)]$ from the transformer is passed through a linear softmax classifier $\zeta$ to yield the final prediction.

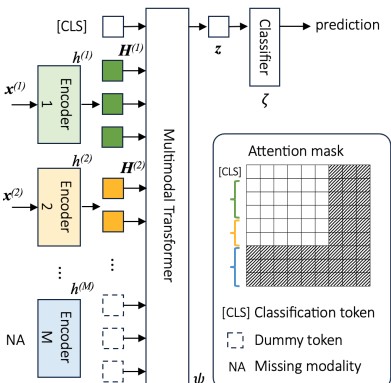

## 3.2 Dynamic Modality Selection at Inference

To address the *discarding-imputation dilemma*, we propose a novel dynamic modality selection algorithm that adaptively discovers valuable task-relevant recovered modalities at inference. The core idea of our approach is a novel reward function that estimates the incremental multimodal task-relevant information contributed by each recovered modality. We establish a theoretical connection between information and task (*i.e.*, classification) loss, leading to an effective reward formulation linked to representation shift

Figure 2: Multimodal network architecture $f$ for arbitrary modalities.

in the latent space. To further enhance robustness, we propose an intra-class similarity calibration for reward refinement based on training data. Finally, we introduce an iterative selection mechanism for reliable dynamic multimodal fusion.

**Multimodal Task-Relevant Information Reward (MTIR)** is designed to estimate the incremental multimodal task-relevant information gained by adding a recovered modality to the existing observed modalities. MTIR is inspired by the notion of information gain (Ma et al., 2019; Jolliffe, 2011) and aims to indicate the marginal impact of each recovered modality on the multimodal representation: (1) *Positive reward* suggests that the recovered modality introduces additional task-relevant information that enhances the representation; (2) *Zero reward* indicates that the recovery is of low fidelity, mainly introducing noise and providing negligible benefit; and (3) *Negative reward* implies that the recovery contains task-relevant but semantically inconsistent information, potentially degrading the representation. This formulation enables the identification of both low-fidelity and misaligned modalities, which have been largely overlooked in prior dynamic MDL work.

To quantify task-relevant information contained in the multimodal representation, we consider mutual information between the multimodal representations $\boldsymbol{Z}$ and the target labels $Y$:

$$I(Y; \boldsymbol{Z}) = \mathbb{E}_{p(y, \boldsymbol{z})} \log \frac{p(y, \boldsymbol{z})}{p(y)p(\boldsymbol{z})}. \tag{1}$$

Since the true data distributions are unknown at inference, we propose to derive a lower bound of $I(Y; \boldsymbol{Z})$, estimated using the empirical test-time classification cross-entropy (CE) loss $\hat{\mathcal{L}}_{\text{ce}}$:

$$I(Y; \boldsymbol{Z}) \geq H(Y) - \hat{\mathcal{L}}_{\text{ce}} - G\sqrt{\frac{\ln(1/\delta)}{2|\mathcal{D}|}}, \quad \text{with probability at least } 1 - \delta, \tag{2}$$

where $H(Y)$ is the entropy of the target labels, $G$ is a conservative upper bound on the per-sample CE loss, $|\mathcal{D}|$ is the size of a test dataset, and $\delta \in (0, 1)$ controls the probability of the bound holding (detailed derivation in Appendix A.1). This bound formalizes the intuition that reducing $\hat{\mathcal{L}}_{\text{ce}}$ can increase the lower bound on $I(Y; \boldsymbol{Z})$, thereby potentially increasing the task-relevant information

in $\boldsymbol{Z}$. Motivated by this insight, we propose to use the empirical test-time CE loss decrease as a tractable proxy for information gain, forming the theoretical foundation of our MTIR reward.

Given an incomplete multimodal test sample $\mathbb{X}_j = \{\boldsymbol{x}^{(m)}\}_{m \in \mathcal{I}_j}$ (notations follow Sec. 3.1), we recover the missing modalities as $\tilde{\mathbb{X}}_j = \{\tilde{\boldsymbol{x}}^{(u)}\}_{u \in ([M] \setminus \mathcal{I}_j)}$ using a recovery function $\Upsilon$ (*e.g.*, a VAE). The test-time empirical CE loss of DyMo $g$, which includes a multimodal network $f$ and a dynamic selection algorithm, is then defined as:

$$\hat{\mathcal{L}}_{\text{ce}} = \frac{1}{|\mathcal{D}|} \sum\nolimits_{j=1}^{|\mathcal{D}|} \ell_{\text{ce}} \left[ g_j(\mathbb{X}_j, \tilde{\mathbb{X}}_j), y_j \right]. \tag{3}$$

To minimize $\hat{\mathcal{L}}_{\text{ce}}$, DyMo should adaptively integrate the recovered modalities in $\tilde{\mathbb{X}}$ that reduce per-sample loss $\ell_{\text{ce}}$. Accordingly, for each sample, we define MTIR of $u$-th recovered modality as:

$$\text{R}(\tilde{\boldsymbol{x}}^{(u)}, \mathbb{X}^O) = \ell_{\text{ce}} \left[ f(\mathbb{X}^O), y \right] - \ell_{\text{ce}} \left[ f(\mathbb{X}^O, \tilde{\boldsymbol{x}}^{(u)}), y \right] = -\log p_f(y|\boldsymbol{z}) + \log p_f(y|\boldsymbol{z}^u), \tag{4}$$

where $\mathbb{X}^O$ denotes the observed modalities (initially $\mathbb{X}$), $\ell_{\text{ce}} = -\log p_f(y|\boldsymbol{z})$, $\boldsymbol{z} = \psi[h(\mathbb{X}^O)]$, and $\boldsymbol{z}^u = \psi[h(\mathbb{X}^O, \tilde{\boldsymbol{x}}^{(u)})]$. Since the true label $y$ is unknown at inference, we substitute it with the predicted labels $\hat{y} = \arg\max f(\mathbb{X}^O)$ and $\hat{y}^u = \arg\max f(\mathbb{X}^O, \tilde{\boldsymbol{x}}^{(u)})$. However, such substitution may undermine the reliability of MTIR, especially under mispredictions or overfitting. To mitigate this, inspired by the robustness and generalizability of metric learning (Vinyals et al., 2016; Chen et al., 2020), we further investigate the representation shifts with respect to the training distribution, measured in the latent space. Specifically, we treat classification as a mixture density estimation problem in feature space, where each class corresponds to a component. Suppose equal class prior probability and exponential family distributions, the posterior probability of $y = k$ given $\boldsymbol{z}$ is:

$$p(y = k|\boldsymbol{z}) = \frac{\exp(-d_\phi(\boldsymbol{z}, \boldsymbol{c}_k))}{\sum_{k'=1}^{K} \exp(-d_\phi(\boldsymbol{z}, \boldsymbol{c}_{k'}))}, \; \boldsymbol{c}_k = \frac{1}{\sum_{i=1}^{N} \mathbb{I}[y_i = k]} \sum_{i:y_i=k} \boldsymbol{z}_i, \tag{5}$$

where $d_\phi$ is a Bregman-divergence type distance function (Banerjee et al., 2005) (*e.g.*, squared Euclidean distance), and $\boldsymbol{c}_k$ is the class prototype estimated from the training dataset. The derivation of this expression is provided in Appendix A.2. Substituting Eq. 5 into Eq. 4 yields:

$$\text{R}(\tilde{\boldsymbol{x}}^{(u)}, \mathbb{X}^O) = -\log \frac{\exp(-d_\phi(\boldsymbol{z}, \boldsymbol{c}_{\hat{y}}))}{\sum_{k'=1}^{K} \exp(-d_\phi(\boldsymbol{z}, \boldsymbol{c}_{k'}))} + \log \frac{\exp(-d_\phi(\boldsymbol{z}^u, \boldsymbol{c}_{\hat{y}^u}))}{\sum_{k'=1}^{K} \exp(-d_\phi(\boldsymbol{z}^u, \boldsymbol{c}_{k'}))}. \tag{6}$$

This formula indicates that higher MTIR rewards can be obtained when the representation moves closer to the class prototype after incorporating a recovered modality, which is consistent with the intuition that fusing effective information increases the model's predictive certainty (Dai et al., 2023).

**Intra-Class Similarity Calibration:** Eq. 6 defines the MTIR reward based on changes in the distance between a sample representation and its predicted class prototype. A challenging case arises when $\hat{y}$ and $\hat{y}^u$ differ, while $\boldsymbol{z}$ and $\boldsymbol{z}^u$ lie at similar distances from their respective prototypes, yielding a near-zero reward. To address this and enhance the reliability of the MTIR reward, we introduce a novel calibration term $\alpha$, which refines the reward by accounting for how representative a sample is within its predicted class cluster. The calibrated MTIR reward $\text{R}^*$ is:

$$\text{R}^*(\tilde{\boldsymbol{x}}^{(u)}, \mathbb{X}^O) = -\log \frac{\exp(-d_\phi(\boldsymbol{z}, \boldsymbol{c}_{\hat{y}}))}{\sum_{k'=1}^{K} \exp(-d_\phi(\boldsymbol{z}, \boldsymbol{c}_{k'}))} + \alpha \times \log \frac{\exp(-d_\phi(\boldsymbol{z}^u, \boldsymbol{c}_{\hat{y}^u}))}{\sum_{k'=1}^{K} \exp(-d_\phi(\boldsymbol{z}^u, \boldsymbol{c}_{k'}))}. \tag{7}$$

To compute $\alpha$, we first define the intra-class similarity (ICS) score of $\boldsymbol{z}$ for class $k$. Unlike many non-parametric statistical approaches (Hastie et al., 2009) that require computing distances to all training samples in class $k$, we propose an efficient approximation. Specifically, we approximate the distribution of distances from the samples in class $k$ to the class prototype $\boldsymbol{c}_k$ as a truncated normal distribution $d_\phi \sim \mathcal{N}(0, \sigma_k^2), d_\phi > 0$, where $\sigma_k$ is estimated from the training data. The ICS score is then written as follows:

$$\text{ICS}(y = k, \boldsymbol{z}) = \mathbb{P}(d_\phi > d_\phi(\boldsymbol{z}, \boldsymbol{c}_k) | d_\phi > 0) = 2(1 - \Phi(d_\phi(\boldsymbol{z}, \boldsymbol{c}_k))), \tag{8}$$

where $\Phi$ is the cumulative distribution function of the normal distribution. ICS quantifies the representativenss of $\boldsymbol{z}$ within a class cluster, with a higher value indicating closer alignment with the training samples in that cluster.

The calibration term is defined as the ratio between the ICS scores of $\boldsymbol{z}^u$ and $\boldsymbol{z}$: $\frac{\mathrm{ICS}(y=\hat{y}^u, \boldsymbol{z}^u)}{\mathrm{ICS}(y=\hat{y}, \boldsymbol{z})}$. Since $\mathbb{X}^O$ are observed, task-relevant modalities, while $\tilde{\boldsymbol{x}}^{(u)}$ is synthetic and can introduce unreliable information, DyMo should be conservative when $\mathrm{ICS}(y=\hat{y}^u, \boldsymbol{z}^u) > \mathrm{ICS}(y=\hat{y}, \boldsymbol{z})$. To this end, we introduce an asymmetric $\alpha$:

$$\alpha = \begin{cases} 1 & \text{if } \mathrm{ICS}(y=\hat{y}^u, \boldsymbol{z}^u) > \mathrm{ICS}(y=\hat{y}, \boldsymbol{z}), \\ \frac{\mathrm{ICS}(y=\hat{y}^u, \boldsymbol{z}^u)}{\mathrm{ICS}(y=\hat{y}, \boldsymbol{z})} & \text{otherwise,} \end{cases} \qquad (9)$$

which is used to calculate R$^*$ in Eq. 7. Thus, if $\boldsymbol{z}^u$ is less representative than $\boldsymbol{z}$ within its predicted class cluster (*i.e.*, $\alpha < 1$), the second term in R is down-weighted, reducing the calibrated MTIR.

**Iterative Modality Selection:** To improve the reliability of DyMo's dynamic process, we introduce an iterative selection algorithm to maximize multimodal task-relevant information for each sample (Algorithm 1). At each step, given an observed set $\mathbb{X}^O$ and a candidate set $\mathbb{X}^C$, we add the recovered modality with the highest MTIR reward to $\mathbb{X}^O$ while removing all ineffective modalities with non-positive rewards from $\mathbb{X}^C$. This stepwise selection ensures that DyMo incorporates only the most informative modalities, effectively mitigating noise accumulation.

---

**Algorithm 1:** DyMo Adaptive Inference

**Input:** Test dataset $\mathcal{D}$, network $f$ (including $h$, $\psi$, and $\zeta$), and a recovery method $\Upsilon$

**for** each test sample $\mathbb{X} = \{\boldsymbol{x}^{(m)}\}_{m \in \mathcal{I}} \in \mathcal{D}$ **do**
    $\tilde{\mathbb{X}} = \{\tilde{\boldsymbol{x}}^{(u)}\}_{u \in [M] \setminus \mathcal{I}} \leftarrow \Upsilon(\mathbb{X})$ ;         // Recover missing modalities
    $\mathbb{X}^O \leftarrow \mathbb{X}, \quad \mathbb{X}^C \leftarrow \tilde{\mathbb{X}}$ ;         // Initialize observed & candidate modality sets
    **while** $\mathbb{X}^C \neq \emptyset$ **do**
        $r^{*(i)} \leftarrow \mathrm{R}^*(\tilde{\boldsymbol{x}}^{(i)}, \mathbb{X}^O), \forall \tilde{\boldsymbol{x}}^{(i)} \in \mathbb{X}^C$ ;         // Calibrated MTIR reward (Eq. 7)
        $k \leftarrow \arg\max_i r^{*(i)}$ ;         // Best candidate modality index
        **if** $r^{*(k)} > 0$ **then**
            $\mathbb{X}^O \leftarrow \mathbb{X}^O \cup \tilde{\boldsymbol{x}}^{(k)}$ ;         // Integrate the most informative modality
        **end**
        $\mathbb{X}^C \leftarrow \mathbb{X}^C \setminus \{\tilde{\boldsymbol{x}}^{(i)} \mid r^{*(i)} \leq 0\}$ ;         // Remove ineffective modalities
    **end**
    $\hat{y} \leftarrow f(\mathbb{X}^O)$
**end**
**return** All test predictions $\hat{\mathcal{Y}}$ collected over $\mathcal{D}$

---

## 3.3 TRAINING ALGORITHM

DyMo's dynamic inference relies on representation shifts, making it essential to train our multimodal network $f$ properly to learn a robust latent feature space where samples of the same class cluster together despite missing modalities. To achieve this, we design incomplete-modality simulation training and an auxiliary missing-agnostic contrastive loss.

**Incomplete Simulation Training.** To guarantee that $f$ extracts robust task-relevant features across various missing patterns, we propose a simple yet effective random sampling strategy during training. In specific, each complete multimodal sample $\{\boldsymbol{x}^{(1)}, ..., \boldsymbol{x}^{(M)}\}$ has $2^M - 1$ non-empty modality subsets. In each minibatch, we randomly sample $A$ such subsets for classification loss calculation:

$$\mathcal{L}_{\text{class}} = -\frac{1}{A} \frac{1}{B} \sum_{\mathcal{S} \sim \mathcal{U}_A} \sum_{i=1}^B \log p_f \left( y_i \mid \{\boldsymbol{x}^{(m)}\}_{m \in \mathcal{S}} \right), \qquad (10)$$

where $B$ is the batch size, and $\mathcal{U}_A \sim \mathrm{Uniform}_{\text{w/o rep}}(\mathcal{P}([M]) \setminus \emptyset, A)$. $\mathcal{P}([M])$ denotes a collection of all modality subsets. This strategy reduces computational cost compared to prior work (Ma et al., 2022) that requires all missing patterns in each training step.

**Auxiliary Missing-Agnostic Contrastive Loss** is designed to further enhance intra-class clustering and inter-class separation regardless of missing scenarios:

$$\mathcal{L}_{\text{aux}} = -\frac{1}{A} \frac{1}{B} \sum_{\mathcal{S} \sim \mathcal{U}_A} \sum_{i=1}^B \log \frac{\exp\left(-d_\phi(\boldsymbol{z}_i, \boldsymbol{c}_{y_i})/t\right)}{\sum_{k'=1}^K \exp\left(-d_\phi(\boldsymbol{z}_i, \boldsymbol{c}_{k'})/t\right)}, \qquad (11)$$

where $t$ is the temperature parameter. In experiments, we test two common distance functions: the squared Euclidean distance $d_\phi(\boldsymbol{u}, \boldsymbol{v}) = \|\boldsymbol{u} - \boldsymbol{v}\|_2^2$ and the cosine distance $d_\phi(\boldsymbol{u}, \boldsymbol{v}) = 1 - (\boldsymbol{u} \cdot \boldsymbol{v})/(\|\boldsymbol{u}\|_2 \|\boldsymbol{v}\|_2)$, where $\boldsymbol{u} \cdot \boldsymbol{v}$ is the dot product of the vectors.

**Overall Loss.** The final training loss for DyMo is expressed as $\mathcal{L}_{\text{overall}} = \mathcal{L}_{\text{class}} + \mathcal{L}_{\text{aux}}$.

## 4 EXPERIMENT

**Datasets and Evaluation Metrics:** We conduct extensive experiments on 5 different datasets with varied modalities (*i.e.*, image, text, and structured table), including 3 simulated benchmark datasets (Sutter et al., 2021): PolyMNIST, MST, and biomodal CelebA, and two large real-world datasets: a natural image dataset, Data Visual Marketing (DVM) (Huang et al., 2022a), and a medical image dataset, UK Biobank (UKBB) (Sudlow et al., 2015). For UKBB, we focus on two cardiac disease classification tasks: coronary artery disease (CAD) and myocardial infarction, using magnetic resonance (MR) images and disease-related tabular features. Following (Du et al., 2025; Sutter et al., 2020), we report the area under the curve (AUC) for UKBB, and accuracy for the remaining datasets. Dataset details are presented in Appendix B.1.

**Implementation Details:** For the recovery method in DyMo, we leveraged MoPoE (Sutter et al., 2021), a multimodal VAE network, for PolyMNIST, MST, and CelebA. Then, for DVM and UKBB, we used TIP (Du et al., 2024), an image-tabular reconstruction framework. Note that DyMo can be deployed with any recovery method. To ensure a fair comparison, all compared methods used the same modality-specific encoders as DyMo. Models were trained on complete datasets and evaluated under various missing scenarios: (i) For PolyMNIST, we set 5 missing rates $\eta = \{0, 0.2, 0.4, 0.6, 0.8\}$, where each sample randomly misses $\eta \times 100\%$ modalities; (ii) For MST and CelebA, we tested different combinations of missing modalities; (iii) For DVM and UKBB, since TIP is a table reconstruction network, we evaluated both full- and intra-table (*i.e.*, partial within-modality) missing. Specifically, we set 7 missing tabular rates $\gamma = \{0, 0.1, 0.3, 0.5, 0.7, 0.9, 1\}$, where each sample randomly misses $\gamma \times 100\%$ tabular features. Note that DyMo can also handle incomplete modality training, as our multimodal network supports arbitrary modalities. Detailed implementation settings for all models are provided in Appendix B.2.

### 4.1 OVERALL RESULTS

**Comparing Against Multmodal Static / Dynamic Fusion SOTAs:** To assess DyMo's effectiveness in selecting valuable task-relevant recovered modalities at inference, we compared it with SOTA multimodal fusion techniques, including static concatenation-based fusion (**CONCAT**) (Baltrušaitis et al., 2018) and 3 dynamic fusion methods, **QMF** (Zhang et al., 2023), **DynMM** (Xue & Marculescu, 2023), and **PDF** (Cao et al., 2024). All methods were provided with the same set of non-missing and recovered modalities at inference. Notably, prior dynamic methods typically require additional modality-specific branches for modality contribution estimation or multi-stage training, whereas DyMo operates directly on multimodal representations without extra modality-specific parameters and relies on single-stage training. We report DyMo's results using cosine distance ($\text{DyMo}_c$) and squared Euclidean distance ($\text{DyMo}_e$).

As shown in Fig. 3, DyMo achieves substantial improvements on most datasets, especially under severe missing scenarios. For example, DyMo achieves 13.12% higher accuracy on PolyMNIST with 80% missing modalities and 4.11% higher accuracy on DVM when the full table is missing. Both $\text{DyMo}_c$ and $\text{DyMo}_e$ consistently outperform prior SOTAs on most datasets, demonstrating DyMo's robustness to the choice of distance metric. We further observe that (1) performance decreases vary across different missing modality combinations (Fig. 3(b,c)), indicating the existence of varying modality task relevance; (2) prior dynamic methods outperform static fusion on DVM (Fig. 3(d)) but achieve limited gains on the 3 simulated benchmarks (Fig. 3(a-c)). This limitation arises because VAE-based reconstruction tends to produce visually plausible but class-misaligned recovered modalities, which prior dynamic methods struggle to handle. By contrast, DyMo addresses this via estimating incremental multimodal task-relevant information gain, resulting in superior results. Additionally, DyMo and CONCAT perform similarly on CAD, likely due to the consistent informativeness of the recovered table modality across samples, leaving limited room for dynamic fusion to further improve performance.

**Comparing Against Incomplete MDL SOTAs:** To evaluate DyMo's efficacy in handling missing modalities, we compared it with SOTA incomplete MDL approaches, including 5 recovery-based (**MultiAE** (Ngiam et al., 2011), **MoPoE** (Sutter et al., 2021), **M3Care** (Zhang et al., 2022), **OnlineMAE** (Woo et al., 2023), and **CMVAE** (Palumbo et al., 2024)) and 4 recovery-free methods (**ModDrop** (Neverova et al., 2015), **MTL** (Ma et al., 2022), **MAP** (Lee et al., 2023), and **MUSE** (Wu et al., 2024b)). For models without simulating different missing scenarios during training, we ad-

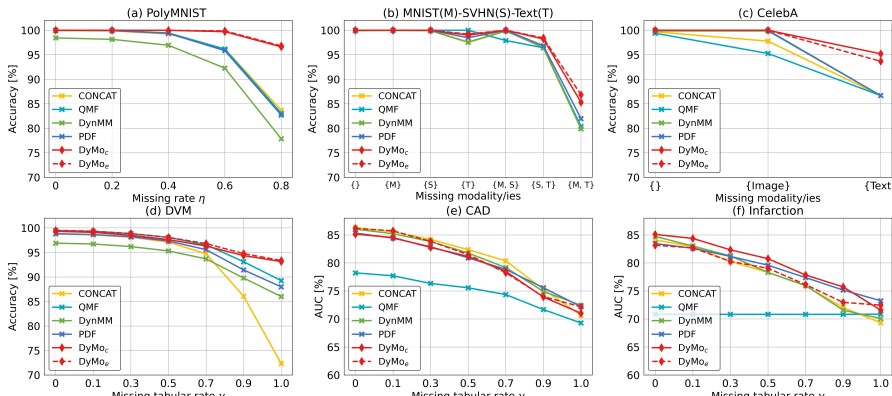

Figure 3: Comparison of DyMo with static/dynamic multimodal fusion techniques on 6 multimodal classification tasks under various missing scenarios. DyMo$_c$ and DyMo$_e$ denote the use of cosine and squared Euclidean distances, respectively.

Table 1: Comparison of DyMo with recovery-based and recovery-free incomplete MDL methods on 6 multimodal classification tasks under various missing scenarios. Models marked with $\dagger$ were trained using our proposed incomplete modality simulation. Complete results for all missing scenarios are provided in Appendix C.1.

| Model | PolyMNIST Acc (%) ↑ | | | MST Acc (%) ↑ | | | CelebA Acc (%) ↑ | | | DVM Acc (%) ↑ | | | CAD AUC (%) ↑ | | | Infarction AUC (%) ↑ | | |
|---|---|---|---|---|---|---|---|---|---|---|---|---|---|---|---|---|---|---|
| | Missing Rate $\eta$ | | | Missing Modality/ies | | | | | | Missing Tabular Rate $\gamma$ | | | | | | | | |
| | 0 | 0.6 | 0.8 | {} | {S,T} | {M,T} | {} | {I} | {T} | 0 | 0.7 | 1 | 0 | 0.7 | 1 | 0 | 0.7 | 1 |
| (a) Recovery-based Methods for Missing Modality | | | | | | | | | | | | | | | | | | |
| MultiAE | 99.77 | 95.36 | 84.39 | 99.96 | 97.00 | 81.60 | 89.98 | 15.51 | 89.71 | - | - | - | - | - | - | - | - | - |
| MultiAE$^\dagger$ | 99.94 | 97.50 | 89.86 | 99.87 | 98.33 | 83.44 | 88.66 | 72.04 | 87.44 | - | - | - | - | - | - | - | - | - |
| MoPoE | 99.79 | 93.93 | 79.84 | 99.62 | 90.86 | 79.01 | 38.97 | 13.91 | 34.84 | - | - | - | - | - | - | - | - | - |
| MoPoE$^\dagger$ | 99.63 | 96.81 | 87.06 | 99.39 | 96.50 | 82.54 | 68.22 | 56.90 | 65.75 | - | - | - | - | - | - | - | - | - |
| M3Care | 99.93 | 56.66 | 40.53 | 99.99 | 16.03 | 9.34 | 92.33 | 99.92 | 51.75 | 98.44 | - | 11.92 | 85.62 | - | 64.99 | 70.61 | - | 70.53 |
| M3Care$^\dagger$ | 99.99 | 97.27 | 87.92 | 99.98 | 98.27 | 85.16 | 98.73 | 97.14 | 91.32 | 98.94 | - | 93.43 | 72.48 | - | 72.48 | 83.27 | - | 68.44 |
| OnlineMAE | 100 | 98.29 | 90.09 | 99.90 | 98.14 | 84.14 | 86.67 | 86.67 | 86.67 | 90.92 | - | 89.90 | 85.22 | - | 70.96 | 84.05 | - | 61.39 |
| CMVAE | 94.93 | 95.13 | 95.11 | - | - | - | - | - | - | - | - | - | - | - | - | - | - | - |
| CMVAE$^\dagger$ | 94.59 | 95.17 | 95.20 | - | - | - | - | - | - | - | - | - | - | - | - | - | - | - |
| (b) Recovery-free Methods for Missing Modality | | | | | | | | | | | | | | | | | | |
| ModDrop | 99.97 | 97.66 | 88.44 | 100 | 98.21 | 82.47 | 99.93 | 99.93 | 87.32 | 99.02 | 93.23 | 87.97 | 85.10 | 76.65 | 69.18 | 84.76 | 74.64 | 72.16 |
| MTL | 99.97 | 98.43 | 91.14 | 99.96 | 98.60 | 84.37 | 99.69 | 99.26 | 89.38 | 99.44 | 95.53 | 92.32 | 84.87 | 77.72 | 70.23 | 83.59 | 75.73 | 69.90 |
| MAP | 99.86 | 43.00 | 23.19 | 100 | 9.83 | 10.13 | 99.98 | 99.93 | 85.33 | 98.86 | 86.96 | 63.15 | 84.39 | 75.61 | 70.11 | 84.62 | 71.73 | 69.17 |
| MAP$^\dagger$ | 99.99 | 96.74 | 76.20 | 99.99 | 97.84 | 11.36 | 100 | 99.99 | 86.06 | 99.37 | 94.83 | 91.15 | 85.26 | 75.84 | 68.76 | 85.49 | 75.88 | 70.81 |
| MUSE | 99.93 | 94.73 | 77.56 | 99.86 | 97.14 | 35.96 | 99.93 | 99.86 | 88.35 | 96.86 | - | 53.23 | 83.47 | - | 53.23 | 84.40 | - | 66.78 |
| (c) Dynamic Recovery Method for Missing Modality | | | | | | | | | | | | | | | | | | |
| DyMo$_c$ | 100 | 99.71 | 96.61 | 100 | 98.22 | 85.31 | 100 | 100 | 95.20 | 99.30 | 96.31 | 93.14 | 85.14 | 78.49 | 71.02 | 85.10 | 77.85 | 71.58 |
| DyMo$_e$ | 100 | 99.87 | 96.81 | 100 | 98.43 | 86.84 | 100 | 100 | 93.67 | 99.50 | 96.81 | 93.36 | 86.17 | 78.24 | 72.17 | 83.16 | 76.14 | 72.47 |

ditionally report their results with our incomplete simulation training. Since ModDrop is a training scheme rather than a standalone architecture, we applied it to the same multimodal backbone as DyMo for a fair comparison. Methods requiring task-specific decoders or restricted to full-modality missing settings were evaluated only on datasets where such decoders are provided and under full-modality missingness.

In Tab. 1, integrating our incomplete simulation training improves model performance (w/ *vs.* w/o $\dagger$), demonstrating its effectiveness in learning features robust to missing data. We also observe the *discarding-imputation dilemma*: (i) recovery-free methods suffer large performance drops when highly task-relevant modalities are missing, *e.g.*, a 61.18% accuracy reduction for MUSE on MST with missing {M,T} *vs.* {S,T} (note that CMVAE is relatively stable across missing rates, as it performs classification using a single randomly selected observed modality); (ii) recovery-based methods struggle under severe missing scenarios, *e.g.*, OnlineMAE's accuracy on PolyMNIST decreases by 9.91% with $\eta = 0.8$ *vs.* $\eta = 0$, indicating the generation of unreliable recoveries. In contrast, DyMo effectively addresses this dilemma, significantly outperforming prior SOTAs on full- and intra-modality missing conditions, *e.g.*, 5.67% higher accuracy on PolyMNIST with 80% missing modalities and 1.97% higher AUC on Infarction with 70% missing tabular features. Per-

Table 2: Ablation study of DyMo. Baseline integrates all recovered modalities without selection. $S$ integrates all modalities with positive reward ($r > 0$) *simultaneously*. $I$ *iteratively* adds the modality with the highest $r$. $C$ uses the calibrated reward $r^*$, obtained via intra-class similarity calibration.

| Dynamic selection | $C$ | PolyMNIST | | MST | | CelebA | | DVM | | CAD | | Infarction | |
|---|---|---|---|---|---|---|---|---|---|---|---|---|---|
| | | 0.6 | 0.8 | {S,T} | {M,T} | {I} | {T} | 0.7 | 1 | 0.7 | 1 | 0.7 | 1 |
| Baseline | | 96.80 | 84.21 | 96.43 | 80.73 | **100** | 86.67 | 96.27 | 88.36 | 79.15 | 71.84 | 78.31 | 74.89 |
| $S$ | | 99.59 | 94.33 | 97.21 | 82.08 | **100** | 86.67 | 96.54 | 92.83 | **79.44** | 72.20 | 77.83 | 75.27 |
| $I$ | | 99.60 | 94.50 | 97.26 | 82.12 | **100** | 86.67 | 96.54 | 92.83 | **79.44** | 72.20 | 77.83 | 75.27 |
| $I$ | ✓ | **99.71** | **96.61** | **98.22** | **85.31** | **100** | **95.20** | 96.31 | **93.14** | 78.49 | 71.02 | **77.85** | 71.58 |

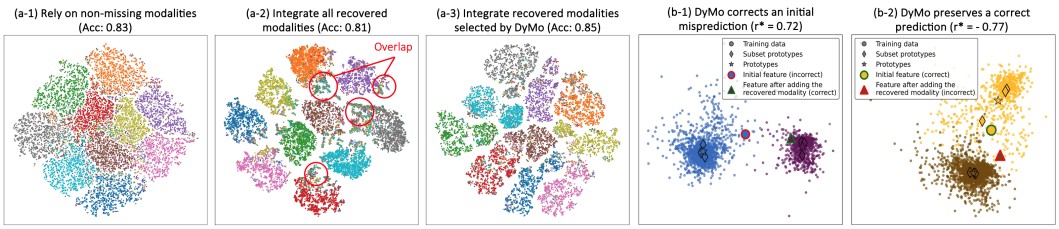

Figure 4: (a) t-SNE visualization of DyMo$_c$ on MST with different modality inputs: (a-1) using only non-missing modalities; (a-2) integrating all recovered modalities without selection; (a-3) incorporating recovered modalities selected by DyMo$_c$. (b) PCA visualizations of two successful DyMo$_c$'s test cases on DVM: (b-1) a misprediction corrected by incorporating a recovered modality; (b-2) a correct prediction maintained by disregarding an unreliable recovered modality.

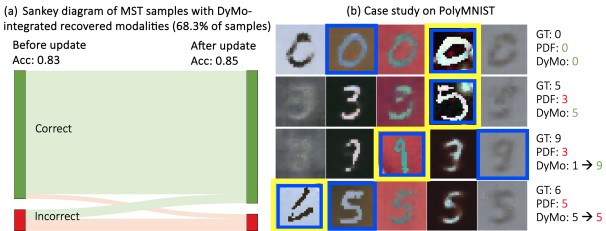

Figure 5: (a) Sankey diagram for DyMo$_c$ prediction transitions on MST with missing {M,T}. (b) Case study on PolyMNIST, where yellow indicates non-missing modalities, while blue indicates modalities selected by DyMo$_c$.

Table 3: Classification accuracy (%) of DyMo$_c$ (firs three rows) and DyMo$_e$ (last three rows) using different modality recovery methods on PolyMNIST under various missing modality rates $\eta$.

| $\eta$ | 0.2 | 0.4 | 0.6 | 0.8 |
|---|---|---|---|---|
| MoPoE | **100** | 99.99 | 99.71 | 96.61 |
| MMVAE+ | 99.99 | 99.97 | 99.79 | 97.40 |
| CMVAE | **100** | 99.97 | **99.80** | **97.69** |
| MoPoE | **100** | 99.99 | 99.87 | 96.81 |
| MMVAE+ | **100** | **100** | 99.87 | 97.39 |
| CMVAE | **100** | 99.99 | **99.89** | 97.39 |

formance matches M3Care$^\dagger$ on DVM and CAD when $\eta = 1$, likely due to TIP's limited full-table reconstruction; stronger recovery could further enhance results.

## 4.2 ABLATION STUDY AND VISUALIZATION

**Effectiveness of Key Model Components:** We ablated the key components of DyMo in Tab. 2, including the MTIR reward, iterative selection, and the calibration term. The results show that integrating all recovered modalities without selection can introduce task-irrelevant information and degrade performance. Each component contributes positively, and DyMo, which combines all of them, achieves the best overall results. For CAD and Infarction, $I$ outperforms $I+C$. This is likely because the calibration term, bounded between 0 and 1, makes the model more conservative in modality selection. In these cases, the recovered table consistently provides task-relevant information, so omitting $C$ allows more samples to benefit from it. Tuning a dataset-specific scaling hyper-parameter before applying $C$ may mitigate this issue, which we leave for future work. Additional ablation studies on our incomplete simulation training, generalizability analysis of DyMo, and adaptive inference analysis can be found in Appendix C.2-C.3. Test-time task loss analysis is reported in Appendix. C.4.

**Robustness of DyMo Across Different Modality Recovery Methods:** We conducted incomplete multimodal experiments using 3 different modality recovery approaches, including MoPoE, MM-

VAE+ (Palumbo et al., 2023) and CMVAE (Palumbo et al., 2024), and quantitatively evaluated their reconstruction performance on PolyMNIST. As shown in Tab. 3, DyMo combined with any of these recovery methods consistently outperforms prior SOTA dynamic/incomplete MDL methods (see Fig. 3 and Tab. 1), despite the varying reconstruction quality of CMVAE and MoPoE (Fig. S6 in Appendix C.6). To further assess DyMo's robustness to reconstruction quality, we additionally evaluated an alternative machine-learning-based recovery method on DVM and UKBB, and conducted an extreme simulation experiment with a controlled correct recovery rate. The results and detailed analyses are provided in Appendix. C.2.

**Visualization of Multimodal Representations and Case Studies:** To examine the effect of integrating recovered modalities selected by DyMo in shaping the latent space, we used t-SNE (Maaten & Hinton, 2008) to visualize the multimodal embeddings of the test set. Fig. 4(a) shows that integrating all recovered modalities increases inter-class separation; however, samples with unreliable recoveries can be embedded within incorrect class clusters, leading to misclassifications and degraded performance. In contrast, DyMo's dynamic fusion alleviates this issue, producing a more discriminative latent space and improved classification results. Moreover, we conducted case studies at both the feature level (PCA (Jolliffe, 2011) visualization, Fig. 4(b)) and the input level (Fig. 5(b)), illustrating that DyMo effectively selects reliable, task-relevant recovered modalities that enhance model performance. A challenging example is shown in the fourth row of Fig. 5(b), where all dynamic methods struggle due to noisy observed modalities and semantically misaligned recovered modalities. Additional visualizations and detailed analyses are provided in Appendix C.5.

**Prediction Transition:** We analyzed prediction changes before and after dynamic modality selection using a Sankey diagram. As shown in Fig. 5(a), DyMo corrects many initial mispredictions and achieves improved performance, highlighting the benefit of recovered modalities. A small fraction of correct predictions become incorrect after updating, likely due to limited recovery quality, suggesting that stronger recovery methods could reduce such errors. Reconstruction analyses of modality recovery methods are in Appendix C.6.

## 5 CONCLUSION

In conclusion, we present the first study on dynamic multimodal fusion after modality recovery for addressing missing modalities. We propose DyMo, a new inference-time dynamic modality selection framework that fully explores task-relevant information while addressing the *discarding-imputation dilemma* by adaptively identifying and fusing valuable recovered modalities. DyMo introduces a novel selection algorithm grounded in maximizing multimodal task-relevant information and proposes a principled reward function. We further design a flexible multimodal network architecture and a tailored training strategy to enable robust multimodal representation learning under arbitrary modality combinations. Experiments on various natural and medical datasets showed DyMo's SOTA performance and the efficacy of its components. With the growing demand for multimodal deep learning, DyMo offers significant potential for real-world deployment on incomplete data. Future work will explore extensions to other tasks (*e.g.*, segmentation) and modalities (*e.g.*, video). An additional discussion for task extension is provided in Appendix. D.

## ACKNOWLEDGEMENTS

This research has been conducted using the UK Biobank Resource under Application Number *40616*.

**Ethics Statement:** This research complies with the ICLR Code of Ethics. The study makes use of multiple datasets: publicly available benchmarks and the UK Biobank (UKBB) Resource under Application Number *40616*. All datasets are de-identified and contain no personally identifiable information. No direct human subject recruitment was involved, and the work does not raise foreseeable risks to participants.

**Reproducibility Statement:** We provide detailed descriptions of datasets, preprocessing steps, model architectures, training procedures, and evaluation protocols in Sec. 3 and Sec. 4 of the

manuscript and Appendix B to ensure reproducibility of our model and compared approaches. Source code and model weights are available at `https://github.com//siyi-wind/DyMo`.

**LLM Usage:** We note that large language models (LLMs) were used solely to improve language clarity; all scientific content and ideas were developed by the authors.

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

# Appendices

**Overview:** The appendices are structured to provide additional details and supporting evidence for the main manuscript. In Appendix A, we provide the detailed formulations of the proposed DyMo. Appendix B describes the datasets used in our experiments, together with the implementation details for both DyMo and the baseline algorithms to ensure reproducibility. Appendix C presents additional experimental results, including complete results under various missing-data scenarios, extended ablation studies, generalization analyses (*e.g.*, additional robustness evaluation for reconstruction quality), adaptive inference analysis, test-time task loss analysis, additional visualizations, and reconstruction performance assessments of modality recovery approaches. Finally, Appendix. D briefly discusses potential extensions of our DyMo beyond classification tasks.

## A DETAILED FORMULATION

### A.1 RELATIONSHIP BETWEEN TASK LOSS FUNCTION & TASK-RELEVANT INFORMATION

Our dynamic selection algorithm leverages a novel multimodal task-relevant information reward (MTIR) to rank the incremental multimodal task-relevant information gained from adding each recovered modality. Since the underlying data distribution is unknown at inference time, we introduce a tractable proxy based on the task (*i.e.*, classification) loss function. In this section, we derive the connection between task loss and task-relevant information, demonstrating that reducing the task loss can increase the lower bound on task-relevant information.

Let the true conditional distribution of labels given representations be $p(y|\boldsymbol{z})$, and the model-predicted distribution be $q_\theta(y|\boldsymbol{z})$, where $\theta$ denotes model parameters, $y$ is the classification label, and $\boldsymbol{z}$ is the multimodal representation. The true classification cross-entropy (CE) loss over a test dataset $\mathcal{D}$ is:

$$\begin{aligned}
\mathcal{L}_{\text{ce}} &= \text{CE}(p(y|\boldsymbol{z}); q_\theta(y|\boldsymbol{z})) \\
&= -\int_{\mathcal{Z}} p(\boldsymbol{z}) \left[ \int_{\mathcal{Y}} p(y|\boldsymbol{z}) \log q_\theta(y|\boldsymbol{z}) dy \right] d\boldsymbol{z}.
\end{aligned} \tag{S1}$$

We can decompose the cross-entropy into conditional entropy and conditional KL divergence:

$$\begin{aligned}
\mathcal{L}_{\text{ce}} &= -\int_{\mathcal{Z}} p(\boldsymbol{z}) \left[ \int_{\mathcal{Y}} p(y|\boldsymbol{z}) \log q_\theta(y|\boldsymbol{z}) dy \right] d\boldsymbol{z} + \int_{\mathcal{Z}} p(\boldsymbol{z}) \left[ \int_{\mathcal{Y}} p(y|\boldsymbol{z}) \log p(y|\boldsymbol{z}) dy \right] d\boldsymbol{z} - \int_{\mathcal{Z}} p(\boldsymbol{z}) \left[ \int_{\mathcal{Y}} p(y|\boldsymbol{z}) \log p(y|\boldsymbol{z}) dy \right] d\boldsymbol{z} \\
&= -\int_{\mathcal{Z}} p(\boldsymbol{z}) \left[ \int_{\mathcal{Y}} p(y|\boldsymbol{z}) \log p(y|\boldsymbol{z}) dy \right] d\boldsymbol{z} + \int_{\mathcal{Z}} p(\boldsymbol{z}) \left[ \int_{\mathcal{Y}} p(y|\boldsymbol{z}) \log \frac{p(y|\boldsymbol{z})}{q_\theta(y|\boldsymbol{z})} \right] d\boldsymbol{z} \\
&= H(Y|\boldsymbol{Z}) + \text{KL}(p(y|\boldsymbol{z})||q_\theta(y|\boldsymbol{z})).
\end{aligned} \tag{S2}$$

Since conditional KL divergence is non-negative (Cover, 1999), we have

$$\mathcal{L}_{\text{ce}} \geq H(Y|\boldsymbol{Z}). \tag{S3}$$

On the other hand, the task-relevant information contained in multimodal representations $\boldsymbol{Z}$ can be quantified by the mutual information between $\boldsymbol{Z}$ and the labels $Y$ (Tishby & Zaslavsky, 2015):

$$\begin{aligned}
I(Y; \boldsymbol{Z}) &= \mathbb{E}_{p(y,\boldsymbol{z})} \log \frac{p(y, \boldsymbol{z})}{p(y)p(\boldsymbol{z})} \\
&= -\mathbb{E}_{p(y,\boldsymbol{z})} \log p(y) + \mathbb{E}_{p(y,\boldsymbol{z})} \log p(y|\boldsymbol{z}) \\
&= H(Y) - H(Y|\boldsymbol{Z}),
\end{aligned} \tag{S4}$$

where $H(Y)$ is the entropy of labels. Combining this with Eq. S3 yields a lower bound on mutual information:

$$I(Y; \boldsymbol{Z}) \geq H(Y) - \mathcal{L}_{\text{ce}} \tag{S5}$$

This shows that the lower bound on mutual information increases as the expected CE loss $\mathcal{L}_{\text{ce}}$ decreases, motivating the use of the loss as a tractable proxy for task-relevant information. In practice,

since the true distributions $p(z)$ and $p(y|z)$ are unknown, we estimate the bound using the empirical test-time CE loss computed from one-hot labels. The empirical test-time CE loss is written as:

$$\hat{\mathcal{L}}_{\text{ce}} = \frac{1}{|\mathcal{D}|} \sum_{j=1}^{|\mathcal{D}|} \ell_{\text{ce}_j} = \frac{1}{|\mathcal{D}|} \sum_{j=1}^{|\mathcal{D}|} -\log q_\theta(y_j|z_j). \tag{S6}$$

Assuming that the model is well-trained, we follow the common bounded-loss assumption in prior work that the per-sample test-time CE loss is bounded in practice (Cao et al., 2024; Zhang et al., 2023; Mohri et al., 2018). Therefore, we adopt a conservative bound:

$$0 \leq \ell_{\text{ce}_j} = -\log q_\theta(y_j|z_j) \leq G, \quad \forall j \in \{1, 2, \ldots, |\mathcal{D}|\}. \tag{S7}$$

Then, by Hoeffding's inequality (Hoeffding, 1963), with probability at least $1 - \delta$,

$$\mathcal{L}_{\text{ce}} \leq \hat{\mathcal{L}}_{\text{ce}} + G\sqrt{\frac{\ln(1/\delta)}{2|\mathcal{D}|}}. \tag{S8}$$

Substituting this into Eq. S5, we obtain a high-probability lower bound on the mutual information in terms of the empirical test-time CE loss:

$$I(Y; \mathbf{Z}) \geq H(Y) - \hat{\mathcal{L}}_{\text{ce}} - G\sqrt{\frac{\ln(1/\delta)}{2|\mathcal{D}|}}, \quad \text{with probability at least } 1 - \delta. \tag{S9}$$

In this equation, the last term $G\sqrt{\frac{\ln(1/\delta)}{2|\mathcal{D}|}}$ is constant and will be nearly to zero when the test dataset size is sufficient large. Therefore, reducing the empirical test-time CE loss can increase the lower bound on task-relevant information contained in multimodal representations $z$, which provides the theoretical foundation of our MTIR reward in the dynamic modality selection algorithm.

## A.2 FORMULATION ON THE FEATURE SPACE

A regular Bregman divergence (Banerjee et al., 2005) $d_\phi$ is defined as:

$$d_\phi(z, z') = \phi(z) - \phi(z') - (z - z')^T \nabla_\phi(z'), \tag{S10}$$

where $\phi$ is a differentiable, strictly convex function of the Legendre type, *e.g.*, squared Euclidean distance or Mahalanobis distance. Inspired by prototypical networks (Snell et al., 2017), the posterior probability $p(y|z)$ can be interpreted as a mixture density estimation on the training set with an exponential family density defined in the feature space.

Any regular exponential family distribution $p_\Psi(z, \theta)$ with natural parameters $\theta$ and cumulant function $\Psi$ can be written in terms of a uniquely determined regular Bregman divergence:

$$p_\Psi(z|\theta) = \exp(\langle z, \theta \rangle - \Psi(\theta))p_0(z) = \exp\left(-d_\phi(z, \boldsymbol{\mu}(\theta))\right)b_\phi(z), \tag{S11}$$

where $b_\phi(z) = \exp(\phi(z))p_0(z)$ and $\boldsymbol{\mu}(\theta)$ is the mean parameter. Consider a regular exponential family mixture model with parameters $\Gamma = \{\theta_k, \pi_k\}_{k=1}^K$:

$$p(z|\Gamma) = \sum_{k=1}^K \pi_k p_\Psi(z|\theta_k) = \sum_{k=1}^K \pi_k \exp\left(-d_\phi(z, \boldsymbol{\mu}(\theta_k))\right)b_\phi(z). \tag{S12}$$

Given $\Gamma$, the posterior probability of assigning an unlabeled data point $z$ to cluster $k$ is:

$$p(y = k|z) = \frac{p(y = k)p(z|y = k)}{p(z|\Gamma)} = \frac{\pi_k \exp(-d_\phi(z, \boldsymbol{\mu}(\theta_k)))}{\sum_{k'=1}^K \pi_{k'} \exp(-d_\phi(z, \boldsymbol{\mu}(\theta_{k'})))} \tag{S13}$$

Following Snell et al. (2017), we assume an equally-weighted mixture model with one cluster per class, where $\pi_k = \frac{1}{K}$, and $\boldsymbol{\mu}(\theta_k) = c_k$ denotes the class prototype estimated from the training data. The posterior probability then simplifies to:

$$p(y = k|z) = \frac{\exp(-d_\phi(z, c_k))}{\sum_{k'=1}^K \exp(-d_\phi(z, c_{k'}))}, \quad c_k = \frac{1}{\sum_{i=1}^N \mathbb{I}[y_i = k]} \sum_{i:y_i=k} z_i. \tag{S14}$$

During training, we utilize the class prototypes to calculate our auxiliary missing-agnostic contrastive loss $L_{\text{aux}}$. To avoid storing instance embeddings, we maintain the cumulative sum of embeddings and sample counts for each class during training and compute the prototype at the end of each epoch.

Note that the representations of a sample may differ across different modality subsets. To account for this, at inference time, we further define subset-specific class prototypes. In specific, for each class $k \in \{1, ..., K\}$ and each non-empty modality subset $\mathcal{S} \subseteq [M], \mathcal{S} \neq \emptyset$, we construct:

$$\boldsymbol{c}_{k,\mathcal{S}} = \frac{1}{\sum_{i=1}^{N} \mathbb{I}[y_i = k]} \sum_{i:y_i=k} \psi\left(h\left(\{\boldsymbol{x}_i^{(m)}\}_{m \in \mathcal{S}}\right)\right), \tag{S15}$$

where $h$ denotes modality-specific encoders and $\psi$ is the multimodal transformer (notations follow Sec. 3.1). We then aggregate these prototypes into an averaged class prototype by considering all possible non-empty modality subsets:

$$\bar{\boldsymbol{c}}_k = \frac{1}{2^M - 1} \sum_{\mathcal{S} \subseteq [M], \, \mathcal{S} \neq \emptyset} \boldsymbol{c}_{k,\mathcal{S}}. \tag{S16}$$

In practice, $\bar{\boldsymbol{c}}_k$ is substituted for $\boldsymbol{c}_k$ in Eq. S14. For the ICS score calculation of a sample representation $\boldsymbol{z}$, we apply its associated modality subset $\mathcal{S}$ to all training samples and compute Eq. 8 using these subset-specific representations.

In our experiments, we evaluated two common distance metrics: the squared Euclidean distance and the cosine distance. While cosine distance does not strictly belong to the class of Bregman divergences, it is widely used in high-dimensional embedding spaces, where directional similarity is often more informative than vector magnitude (Li et al., 2021; Yang et al., 2023; Du et al., 2025). Our experimental results demonstrate that cosine distance achieves performance comparable to Bregman divergences, suggesting its suitability as a metric for our modality selection algorithm.

## B  DATASET AND IMPLEMENTATION DETAILS

### B.1  DETAILED DATA DESCRIPTION

As summarized in Tab. S1, we conduct extensive experiments on 5 datasets with diverse modalities, *e.g.*, image, text, and structured tables. These comprises of 3 simulated benchmark datasets (Sutter et al., 2021; 2020): PolyMNIST, MNIST-SVHN-Text (MST), and biomodal CelebA (CelebA), as well as two large real-world datasets: a natural image dataset, Data Visual Marketing (DVM) (Huang et al., 2022a) and a medical image dataset, UK Biobank (UKBB) (Sudlow et al., 2015).

PolyMNIST consists of 5 images per data point, all sharing the same digit label but with different backgrounds and handwriting styles. MST is a trimodal dataset combining MNIST, street view house numbers (SVHN), and synthetic text features. CelebA contains facial images paired with descriptive text annotations of facial attributes. The DVM dataset includes 2D RGB images of cars along with tabular data describing vehicle characteristics. Following (Du et al., 2024; Hager et al., 2023; Du et al., 2025), we used 17 tabular features, including 4 categorical features (*e.g.*, color), and 13 continuous features (*e.g.*, width). The UKBB dataset consists of cardiac magnetic resonance images (MRIs) accompanied by tabular data related to cardiovascular diseases. Following prior work (Hager et al., 2023), we used mid-ventricle slices from MRIs at three time points, *i.e.*, end-systolic (ES) frame, end-diastolic (ED) frame, and an intermediate time frame between ED and ES. In addition, we employed 75 disease-related tabular features, including 26 categorical features (*e.g.*, alcohol drinker status) and 49 continuous features (*e.g.*, average heart rate). Notably, due to low disease prevalence, similar to (Du et al., 2025), we constructed 2 balanced training subsets for CAD and Infarction tasks, respectively. Detailed benchmark information for DVM and UKBB, including the complete list of tabular feature names, can be found in the supplementary material of (Du et al., 2024).

### B.2  IMPLEMENTATION DETAILS

For both DVM and UKBB, we adopted the data augmentation strategies described in (Hager et al., 2023; Du et al., 2024). For image data, we applied random scaling, rotation, shifting, flipping, Gaussian noise, as well as brightness, saturation, and contrastive changes, followed by resizing all images

Table S1: Summary of the 5 multimodal datasets for evaluation.

| Dataset | Classification Task | #Modality | Modality Type | #Train | #Val | #Test | #Class |
|---|---|---|---|---|---|---|---|
| PolyMNIST | Digit | 5 | RGB image | 60,000 | 3,000 | 7,000 | 10 |
| MST | Digit | 3 | RGB image, Text | 1,121,360 | 60,000 | 140,000 | 10 |
| CelebA | Face attribute | 2 | RGB image, Text | 162,770 | 19,962 | 19,867 | 2 |
| DVM | Car model | 2 | RGB image, Table | 70,565 | 17,642 | 88,207 | 283 |
| UKBB | Coronary artery disease (CAD) | 2 | MR image, Table | 3,482 | 6,510 | 3,617 | 2 |
| | Myocardial infarction | 2 | MR image, Table | 1,552 | 6,510 | 3,617 | 2 |

Table S2: Hyper-parameter configurations for DyMo.

| Hyper-parameter | PolyMNIST | MST | CelebA | DVM | CAD | Infarction |
|---|---|---|---|---|---|---|
| # layers of the multimodal transformer | 2 | 2 | 2 | 2 | 2 | 2 |
| # attention heads of the multimodal transformer | 4 | 2 | 2 | 8 | 8 | 8 |
| Hidden dimension of the multimodal transformer | 128 | 32 | 32 | 256 | 256 | 256 |
| Sequence length per modality | {4,4,4,4,4} | {1,1,1} | {1,1} | {I:16,T:17} | {I:16,T:75} | {I:16,T:75} |
| Latent space dimension | 64 | 16 | 16 | 128 | 128 | 128 |
| # sampled modality subsets ($A$) | 5 | 2 | 2 | 2 | 2 | 2 |
| Learning rate | 1e-3 | 1e-3 | 1e-3 | 3e-4 | 3e-4 | 3e-4 |
| Batch size | 256 | 256 | 256 | 256 | 128 | 128 |
| Maximum # epochs | 100 | 20 | 20 | 300 | 300 | 300 |

to $128 \times 128$. For tabular data, categorical values (*e.g.*, *yes*, *no*, and *blue*) were converted into ordinal integers, while continuous (numerous) values were standardized using $z$-score normalization. To further enhance data diversity, we randomly replaced 30% of the tabular values for each subject with randomly sampled values from the corresponding columns. Hyper-parameter settings for the proposed DyMo and all compared approaches are detailed below.

**Our DyMo:** For the recovery method in DyMo, we leveraged MoPoE (Sutter et al., 2021), a multimodal VAE, for PolyMNIST, MST, and CelebA. For DVM and UKBB, we used TIP (Du et al., 2024), an image-tabular framework capable of reconstructing tabular features from images and incomplete tables. Importantly, DyMo is agnostic to the choice of recovery method and can be deployed with any recovery method. For modality-specific encoders, as done in (Sutter et al., 2021), we used convolutional neural networks (CNNs) for images and multi-layer perceptrons (MLPs) for text in PolyMNIST, MST, and CelebA. For DVM and UKBB, we followed (Du et al., 2024), using ResNet-50 as the image encoder and a transformer-based encoder for tabular data. The tabular encoder consists of 4 transformer layers, each with 8 attention heads and a hidden dimension of 512. To ensure fairness, all comparing approaches employed the same encoders as DyMo. To mitigate the curse of dimensionality, multimodal representations $z$ were projected into a low-dimensional latent space using a 2-layer MLP before distance computation. The temperature parameter $t$ for distance metrics was set to 0.1. Hyper-parameter configurations for DyMo are summarized in Tab. S2.

**CONCAT (Baltrušaitis et al., 2018):** This static fusion algorithm has been widely adopted in MDL tasks due to its simplicity and effectiveness. CONCAT concatenates modality-specific features into a unified multimodal representation, which is then used for the classification task.

**QMF (Zhang et al., 2023):** This dynamic MDL framework performs modality fusion through uncertainty-aware weighing. Modality-specific uncertainty is estimated via an energy score computed from the output logits of each unimodal network. In addition, QMF incorporates a regularization loss based on the historical training trajectory. Following the original paper, the number of gradient accumulation steps was set to 12 for CAD and Infarction, and 24 for the remaining tasks.

**DynMM (Xue & Marculescu, 2023):** This dynamic MDL approach employs a learnable gating network. Given $M$ modalities, it constructs $M + 1$ network branches: $M$ modality-specific networks and one concatenation-based multimodal network. The gating network adaptively selects one branch for decision-making. Note that this approach requires two-stage training: (i) pre-training each network branch independently on the targe task, and (ii) jointly fine-tuning all network branches together with the gating network.

**PDF (Cao et al., 2024):** This dynamic MDL framework introduces a modality confidence score to fuse unimodal predictions. The score is derived from the predicted class probability of each

unimodal network. In addition, to address potential uncertainty, a relative calibration strategy is further applied to calibrate the confidence scores.

**MultiAE (Ngiam et al., 2011):** This incomplete MDL model is an offline recovery-based method. It first pre-trains a multimodal autoencoder (AE) with a reconstruction objective, then freezes the AE encoder and uses the extracted latent features to train a classifier for downstream tasks. Since the original design does not simulate missing modalities during training, we incorporated our incomplete simulation training to obtain MultiAE$^\dagger$. The hyper-parameter settings of this strategy were aligned with our DyMo (see Tab. S2). The number of pre-training epochs was set to 300 for PolyM-NIST and 200 for MST and CelebA. The same learning rate was used for both pre-training and fine-tuning, matching the learning rate used by DyMo in its one-stage training.

**MoPoE (Sutter et al., 2021):** This incomplete MDL approach is also offline recovery-based. It first pre-trains a multimodal VAE with an ELBO formulation tailored for incomplete multimodal data. The encoder is then frozen, and its extracted latent features are used to train a classifier for downstream tasks. Pre-training was performed for 100 epochs, and the learning rates for pre-training and fine-tuning were identical, equal to that used by DyMo.

**M3Care (Zhang et al., 2022):** This incomplete MDL framework is an online recovery-based method. It imputes missing modality information in the latent space using auxiliary information from similar patents, identified via a task-guided, modality-adaptive similarity metric. Since the original design does not simulate modality missing during training, we incorporated our incomplete simulation training to obtain M3Care$^\dagger$. The hyper-parameter settings of this strategy were aligned with our DyMo.

**OnlineMAE (Woo et al., 2023):** This incomplete MDL model is an online recovery-based method. It jointly optimizes a feature-level reconstruction task and the target classification task by randomly dropping modality features and reconstructing them from the remaining modalities. In reproduction, we observed that training from scratch often led to model collapse, likely because the network initially fails to extract reliable features, making feature-level reconstruction unstable. To mitigate this, we initialized the encoders with weights from MultiAE (after downstream task training) for PolyMNIST, MST, and CelebA, and weights from TIP (after downstream task training) for DVM, CAD, and Infarction.

**ModDrop (Neverova et al., 2015):** This incomplete MDL framework is a missing-agnostic, recovery-free method. It applies random modality dropping during target task training to encourage modality-agnostic representation learning. Each modality is dropped according to a Bernoulli distribution with probability $p$. For a fair comparison, we used the same multimodal network architecture as our DyMo. Following the original paper, $p$ was set to 0.5.

**MTL (Ma et al., 2022):** This incomplete MDL model is a missing-aware, recovery-free method. It introduces a multimodal transformer architecture with missing-aware [CLS] tokens, which encode different missing patterns. Attention masks on those [CLS] tokens ensure that they only aggregate information from observed modalities. To optimize missingness-specific parameters, a multi-task learning strategy is used to compute loss across all missing patterns in each training step. The sequence length per modality and transformer configurations matched those of our DyMo.

**MAP (Lee et al., 2023):** This incomplete MDL approach is also missing-aware and recovery-free. It designs a multimodal transformer architecture with missing-aware prompts. Prompts are assigned based on the missing pattern of each input and injected into multiple transformer blocks. Since the original paper does not simulate missing modalities during training, we applied our incomplete simulation training to obtain MAP$^\dagger$. The sequence length per modality and transformer configurations followed those of our DyMo. Following the original paper, prompt lengths were set to 12 for PolyMNIST, 2 for MST and CelebA, 8 for DVM, and 16 for CAD and Infarction. Prompts were applied to 2 transformer layers.

**MUSE (Wu et al., 2024b):** This incomplete MDL framework is a missing-agnostic, recovery-free method. It models patient-modality relationships using a flexible bipartite graph that supports arbitrary missing-modality patterns. The vertex set includes patient nodes and modality nodes, and edges are defined by the modality missingness matrix. To learn modality-agnostic features, a mutual-consistent contrastive loss is applied, where each edge is dropped with probability $p$. Following the original paper, $p$ was set to 0.5.

Table S3: Complete results on PolyMNIST and MST under various missing scenarios, comparing DyMo with incomplete MDL methods, including both recovery-based and recovery-free approaches. Models marked with † were trained using our proposed incomplete-modality simulation training. In addition to the results reported in Tab. 1, we also include PolyMNIST results for $\eta = \{0.2, 0.4\}$ and MST results for missing modality subsets {M}, {S}, {T}, and {M,S}.

| Model | Publication | PolyMNIST Acc (%) ↑ | | | | | MST Acc (%) ↑ | | | | | | |
|---|---|---|---|---|---|---|---|---|---|---|---|---|---|
| | | Missing Rate $\eta$ | | | | | Missing Modality/ies | | | | | | |
| | | 0 | 0.2 | 0.4 | 0.6 | 0.8 | {} | {M} | {S} | {T} | {M,S} | {S,T} | {M,T} |
| (a) Recovery-based Methods for Missing Modality | | | | | | | | | | | | | |
| MultiAE | ICML'11 | 99.77 | 99.39 | 98.56 | 95.36 | 84.39 | 99.96 | 99.74 | 99.98 | 98.30 | **100** | 97.00 | 81.60 |
| MultiAE† | ICML'11 | 99.94 | 99.83 | 99.03 | 97.50 | 89.86 | 99.87 | 98.65 | 99.93 | 98.60 | **100** | 98.33 | 83.44 |
| MoPoE | ICLR'21 | 99.79 | 99.41 | 98.13 | 93.94 | 79.84 | 99.62 | 91.32 | 99.33 | 92.28 | 90.11 | 90.86 | 79.01 |
| MoPoE† | ICLR'21 | 99.63 | 99.57 | 98.96 | 96.81 | 87.06 | 99.39 | 99.42 | 99.18 | 97.62 | **100** | 96.50 | 82.54 |
| M3Care | KDD'22 | 99.93 | 89.26 | 74.93 | 56.66 | 40.53 | 99.99 | 40.64 | 85.18 | 23.48 | 41.69 | 16.03 | 9.34 |
| M3Care† | KDD'22 | 99.99 | 99.93 | 99.40 | 97.27 | 87.92 | 99.98 | **100** | 99.98 | 99.17 | **100** | 98.27 | 85.16 |
| OnlineMAE | AAAI'23 | **100** | 99.89 | 99.61 | 98.29 | 90.09 | 99.90 | 99.70 | 99.96 | 97.94 | **100** | 98.14 | 84.14 |
| CMVAE | ICLR'24 | 94.93 | 95.50 | 95.21 | 95.13 | 95.11 | - | - | - | - | - | - | - |
| CMVAE† | ICLR'24 | 94.59 | 95.23 | 94.93 | 95.17 | 95.20 | - | - | - | - | - | - | |
| (b) Recovery-free Methods for Missing Modality | | | | | | | | | | | | | |
| ModDrop | ICML'15 | 99.97 | 99.77 | 99.43 | 97.66 | 88.44 | **100** | **100** | **100** | 99.14 | **100** | 98.21 | 82.47 |
| MTL | CVPR'22 | 99.97 | 99.91 | 99.61 | 98.43 | 91.14 | 99.96 | 99.97 | 99.96 | 98.82 | **100** | **98.60** | 84.37 |
| MAP | CVPR'23 | 99.86 | 95.29 | 66.40 | 43.00 | 23.19 | **100** | 65.32 | 58.14 | 15.28 | 80.29 | 9.83 | 10.13 |
| MAP† | CVPR'23 | 99.99 | 99.86 | 99.41 | 96.74 | 76.20 | 99.99 | **100** | 99.99 | 97.87 | **100** | 97.84 | 11.36 |
| MUSE | ICLR'24 | 99.93 | 99.64 | 98.61 | 94.73 | 77.56 | 99.86 | 99.99 | 99.83 | 97.51 | **100** | 97.14 | 35.96 |
| (c) Dynamic Recovery Method for Missing Modality | | | | | | | | | | | | | |
| DyMo$_c$ | | **100** | **100** | **99.99** | 99.71 | 96.61 | **100** | **100** | **100** | 99.01 | **100** | 98.22 | 85.31 |
| DyMo$_e$ | | **100** | **100** | **99.99** | **99.87** | **96.81** | **100** | 99.98 | **100** | **99.22** | **100** | 98.43 | **86.84** |

Table S4: Complete results on DVM, CAD, and Infarction under various missing scenarios, comparing DyMo with incomplete MDL methods, including recovery-based and recovery-free approaches. Models marked with † were trained using our proposed incomplete-modality simulation training. In addition to the results reported in Tab. 1, we also include results for $\gamma = \{0.1, 0.3, 0.5, 0.9\}$.

| Model | DVM Acc (%) ↑ | | | | | | | CAD AUC (%) ↑ | | | | | | | Infarction AUC (%) ↑ | | | | | | |
|---|---|---|---|---|---|---|---|---|---|---|---|---|---|---|---|---|---|---|---|---|---|
| | Missing Tabular Rate $\gamma$ | | | | | | | Missing Tabular Rate $\gamma$ | | | | | | | Missing Tabular Rate $\gamma$ | | | | | | |
| | 0 | 0.1 | 0.3 | 0.5 | 0.7 | 0.9 | 1 | 0 | 0.1 | 0.3 | 0.5 | 0.7 | 0.9 | 1 | 0 | 0.1 | 0.3 | 0.5 | 0.7 | 0.9 | 1 |
| (a) Recovery-based Methods for Missing Modality | | | | | | | | | | | | | | | | | | | | | |
| M3Care | 98.44 | - | - | - | - | - | 11.92 | 85.62 | - | - | - | - | - | 64.99 | 70.61 | - | - | - | - | - | 70.53 |
| M3Care† | 98.94 | - | - | - | - | - | **93.43** | 72.48 | - | - | - | - | - | **72.48** | 83.27 | - | - | - | - | - | 68.44 |
| OnlineMAE | 90.92 | - | - | - | - | - | 89.90 | 85.22 | - | - | - | - | - | 70.96 | 84.05 | - | - | - | - | - | 61.39 |
| (b) Recovery-free Methods for Missing Modality | | | | | | | | | | | | | | | | | | | | | |
| ModDrop | 99.02 | 98.73 | 97.68 | 95.88 | 93.23 | 89.80 | 87.97 | 85.10 | 84.56 | 82.96 | 80.39 | 76.65 | 70.77 | 69.18 | 84.76 | 83.44 | 80.61 | 78.36 | 74.64 | 72.06 | 72.16 |
| MTL | 99.44 | **99.43** | **98.70** | 97.42 | 95.53 | 93.38 | 92.32 | 84.87 | 85.22 | 83.82 | 80.98 | 77.72 | 73.08 | 70.23 | 83.59 | 83.90 | 81.33 | 79.26 | 75.73 | 69.82 | 69.90 |
| MAP | 98.86 | 98.53 | 97.02 | 93.83 | 86.96 | 74.88 | 63.15 | 84.39 | 83.81 | 81.52 | 78.58 | 75.61 | 71.39 | 70.11 | 84.62 | 83.40 | 78.80 | 75.92 | 71.73 | 68.47 | 69.17 |
| MAP† | 99.37 | 99.10 | 98.29 | 96.89 | 94.83 | 92.43 | 91.15 | 85.26 | 84.66 | 82.76 | 80.14 | 75.84 | 70.36 | 68.76 | **85.49** | **84.45** | 82.08 | 78.94 | 75.88 | 71.38 | 70.81 |
| MUSE | 96.86 | - | - | - | - | - | 1.64 | 83.47 | - | - | - | - | - | 53.23 | 84.40 | - | - | - | - | - | 66.78 |
| (c) Dynamic Recovery Method for Missing Modality | | | | | | | | | | | | | | | | | | | | | |
| DyMo$_c$ | 99.30 | 99.10 | 98.50 | 97.62 | 96.31 | 94.36 | 93.14 | 85.14 | 84.53 | 82.77 | 81.16 | **78.49** | 73.92 | 71.02 | 85.10 | 84.38 | **82.34** | **80.76** | **77.85** | **75.72** | 71.58 |
| DyMo$_e$ | **99.50** | 99.33 | 98.86 | **98.08** | **96.81** | **94.76** | 93.36 | **86.17** | **85.73** | **83.84** | **81.49** | 78.24 | **74.01** | 72.17 | 83.16 | 82.70 | 80.27 | 79.05 | 76.14 | 72.96 | **72.47** |

We trained all models using the Adam optimizer (Kingma & Ba, 2014) without weight decay and ran experiments on a single NVIDIA A5000 GPU. To mitigate overfitting, similar to (Du et al., 2024; Hager et al., 2023), we employed an early stopping strategy, with a minimal divergence threshold of 0.0001, a maximal number of training epochs (see Tab. S2), and a patience (stopping threshold) of 20 epochs. All models were trained with the same learning rate and batch size as our DyMo (see Tab. S2). We ensured convergence of all methods under this configuration.

# C  ADDITIONAL EXPERIMENT

## C.1  COMPARING AGAINST INCOMPLETE MDL SOTAS (COMPLETE RESULTS)

In Sec. 4.1 of the manuscript, we report results for the most challenging missing scenarios on PolyMNIST, MST, DVM, CAD, and Infarction in Tab. 1, due to space constraints. Here, we provide the complete results in Tab. S3 and Tab. S4.

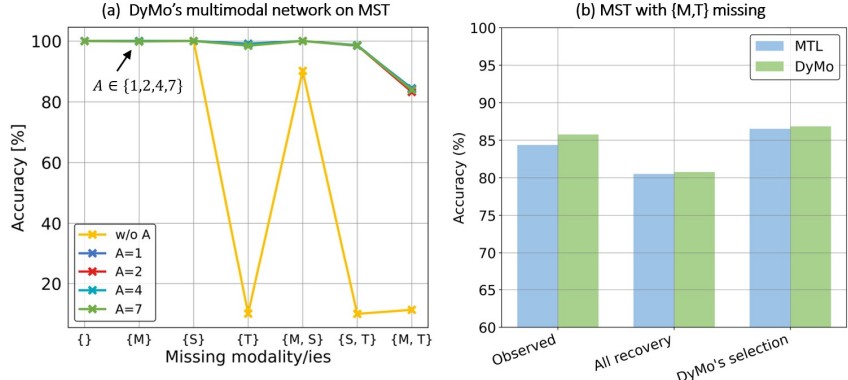

Figure S1: (a) Performance of DyMo$_e$'s multimodal network on MST under various missing scenarios, evaluated with different numbers of sampled modality subsets ($A$). *w/o A* denotes training without our incomplete-modality simulation strategy. (b) Comparison between MTL and DyMo$_e$ on MST with {M,T} missing, under different modality inputs: (1) using only non-missing modalities; (2) integrating all recovered modalities without selection; (3) integrating only the recovered modalities selected by DyMo$_e$.

Table S5: Classification accuracy (%) of DyMo using different modality recovery methods on DVM and Infarction under various missing modality rates.

| Model | Recovery Method | DVM $\eta = 0.7$ | DVM $\eta = 0.9$ | Infarction $\eta = 0.7$ | Infarction $\eta = 0.9$ |
|---|---|---|---|---|---|
| CONCAT | IMI | 81.85 | 67.13 | 73.75 | 69.39 |
| DyMo$_e$ | IMI | 95.97 | 94.28 | 75.96 | 71.09 |
| CONCAT | TIP | 94.74 | 86.08 | 75.91 | 72.04 |
| DyMo$_e$ | TIP | **96.81** | **94.76** | **76.14** | **72.96** |

## C.2  Ablation Study & Generalizability Analysis

**Effect of the Incomplete Simulation Training Strategy:** We examined the impact of the number of sampled modality subsets ($A$) during incomplete-modality simulation training on MST under various missing scenarios. As shown in Fig. S1(a), removing our incomplete simulation training leads to a substantial performance drop, *e.g.*, accuracy decreases by 71.91% on MST with missing {M,T}, compared to models trained with $A = 2$. In contrast, when this training strategy is applied, performance remains stable across different values of $A$, suggesting that even a small $A$ is sufficient to achieve both efficiency and strong performance.

**Efficacy and Applicability of DyMo's Selected Recovered Modalities:** We compared DyMo's multimodal network with MTL, a recovery-free transformer-based method, under three input settings: (1) using only non-missing modalities; (2) integrating all recovered modalities without selection; and (3) integrating the recovered modalities selected by DyMo. As shown in Fig. S1(b), for both models, naively integrating all recovered modalities results in lower accuracy than using only the non-missing modalities, indicating that some imputed modalities are task-irrelevant and can negatively affect decision-making. In contrast, integrating only the recovered modalities selected by DyMo improves performance of both models, demonstrating the effectiveness of DyMo's selection algorithm and the applicability of its selected recovered modalities to other models. Moreover, DyMo consistently outperforms MTL across all input settings, which showcases the efficacy of DyMo's network architecture and training strategy.

**Robustness of DyMo to Modality Recovery (Reconstruction) Quality:** In Sec. 4.2 of the manuscript, we evaluate robustness of DyMo across different modality recovery methods on PolyMNIST. Here, we further assess an alternative recovery method on DVM and UKBB. Specifically, we replace TIP with the iterative multivariate imputer (IMI), a widely-used tabular imputation method that recovers missing values using information from other table columns (Liu et al., 2014). Since IMI

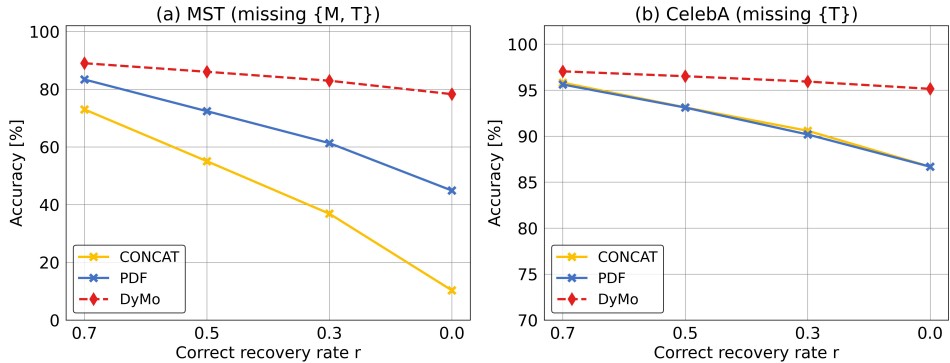

Figure S2: Results on MST (missing {M, T}) and CelebA (missing {T}) with different correct recovery rates $r$.

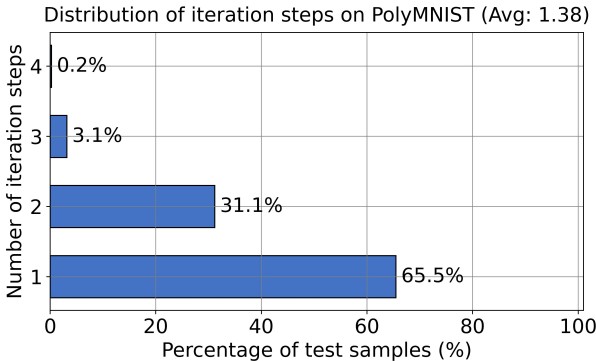

Figure S3: Distribution of iterative selection steps per test sample for DyMo on PolyMNIST with missing modality rate $\eta = 0.8$ (*i.e.*, each sample randomly misses 4 out of 5 modalities). The average number of steps per sample is 1.38.

relies only on table information, it yields lower-quality reconstructions than TIP (Liu et al., 2014). As shown in Tab. S5, with this weaker reconstructor, CONCAT suffers a large accuracy drop compared to its performance with TIP (*e.g.*, a $-18.95\%$ accuracy decrease on DVM with 90% missing tabular features). In contrast, DyMo remains stable and achieves the best performance under both recovery methods. These results show that DyMo is not constrained by the reconstruction-quality bottleneck.

To further stress-test DyMo's under noisy recovery quality, we conducted an extreme simulation experiment with a controlled correct recovery rate $r$. For $r \times 100\%$ of samples, their missing modalities are imputed using their ground-truth versions, while for remaining $(1 - r) \times 100\%$, their missing modalities are replaced with zero-valued noise. We evaluate DyMo on MST (missing M,T) and CelebA (missing T). As shown in Fig. S2, prior static and dynamic fusion methods degrade sharply as recovery quality deteriorates. In contrast, DyMo remains relatively stable across a wide range of correct recovery rates, suggesting its ability to effectively disregard unreliable recovered modalities.

### C.3 ADAPTIVE INFERENCE ANALYSIS

To study how DyMo dynamically selects task-relevant recovered modalities at inference time, we visualized the distribution of iterative selection steps per sample on PolyMNIST, when 80% of modalities are missing. Fig. S3 shows that samples require different numbers of iterations, reflecting variations in the quality of recovered modalities and the adaptive nature of DyMo. At each iteration, only recovered modalities that provide a non-negligible incremental multimodal task-relevant information are added. In addition, most samples require only 1-2 steps, and the average number of

Table S6: Test loss range on PolyMNIST with 80% missing modalities ($\delta = 0.1$).

| Dataset | Dataset Size $|\mathcal{D}|$ | Min | Max ($G$ in Eq. 2) | $G\sqrt{|(\ln 1/\delta)/|\mathcal{D}|}$ in Eq. 2 |
|---|---|---|---|---|
| PolyMNIST | 7,000 | 0.00 | 4.14 | 0.0053 |

iterations is 1.38, suggesting that although iterative selection introduces some additional computation, the extra cost compared to DyMo w/o iteration selection is moderate.

## C.4 TEST-TIME TASK LOSS ANALYSIS

We show the test CE loss range for DyMo on PolyMNIST with 60% missing modalities in Tab. S6. The results show that the test CE loss stays within a well-behaved numerical range. The $G$-related term is extremely small ($< 0.006$), and thus will not affect the tightness of the lower bound in practice (Eq. 2).

## C.5 VISUALIZATION

**Latent Feature Space Visualization on Training Data:** The MTIR reward of a recovered modality in DyMo is based on the representation shift in the latent space relative to the training distribution after adding that modality. We used t-SNE to visualize the multimodal latent space learnt by DyMo's multimodal network on training data under various missing scenarios. As shown in Fig. S4, samples from different classes are generally well-separated, demonstrating that DyMo effectively learns a structured feature space suitable for reward calculation.

**Input-Level Case Study:** To understand which recovered modalities are selected by DyMo, we conduct input-level case studies comparing DyMo with other dynamic/static fusion methods, as shown in Fig. 5(b) in the manuscript and Fig S5. The results indicate that (1) the recovery method may generate modalities of varying qualities across samples; (2) existing dynamic MDL methods that rely solely on modality-specific information for estimating modality importance often fail to identify semantically misaligned recovery (*e.g.*, right case in Fig. S5(b)), which can degrade model performance; and (3) DyMo, however, alleviates this limitation by selectively incorporating beneficial recovery while disregarding unreliable one, thereby improving overall performance. Moreover, Fig. S5(d) illustrates particularly challenging cases for all models, where both non-missing and recovered modalities provide limited task-relevant information. In these rare and difficult cases, DyMo may not fully correct its initial mispredictions, suggesting that the use of more advanced recovery methods could further enhance performance.

## C.6 RECOVERED MODALITY ANALYSIS

**VAE-based Reconstruction Analysis:** Fig. 5(b) in the manuscript and Appendix C.5 present examples of recovered modalities. We further provide a quantitative analysis of these reconstructions. Specifically, We evaluated the reconstruction performance of VAE-based modality recovery methods (*i.e.*, MoPoE, MMVAE+, and CMVAE) on PolyMNIST. Following prior studies (Palumbo et al., 2023; Sutter et al., 2021), we assessed cross-modal generation using two complementary metrics: (i) semantic generative coherence, measured by the accuracy of generated modalities (*i.e.*, conditional coherence accuracy); and (ii) generative quality, measured by the similarity between generated and real samples using the Fréchet Inception Distance (FID) score (Heusel et al., 2017). Additional details of these metrics can be found in Appendix D.3 of (Palumbo et al., 2023). As shown in Fig. S6, CMVAE consistently outperforms the other methods in both cohenrence and quality across different numbers of input modalities. This finding aligns with the observation that DyMo combined with CMVAE achieves the best classification performance (see Tab. 3). Notably, while MoPoE performs substantially worse than MMVAE+ and CMVAE, DyMo exhibits smaller performance gaps across all three recovery methods. This demonstrates that DyMo is robust to different recovery techniques by dynamically integrating task-relevant recovered modalities while disregarding unreliable ones, highlighting its selective and adaptive behavior.

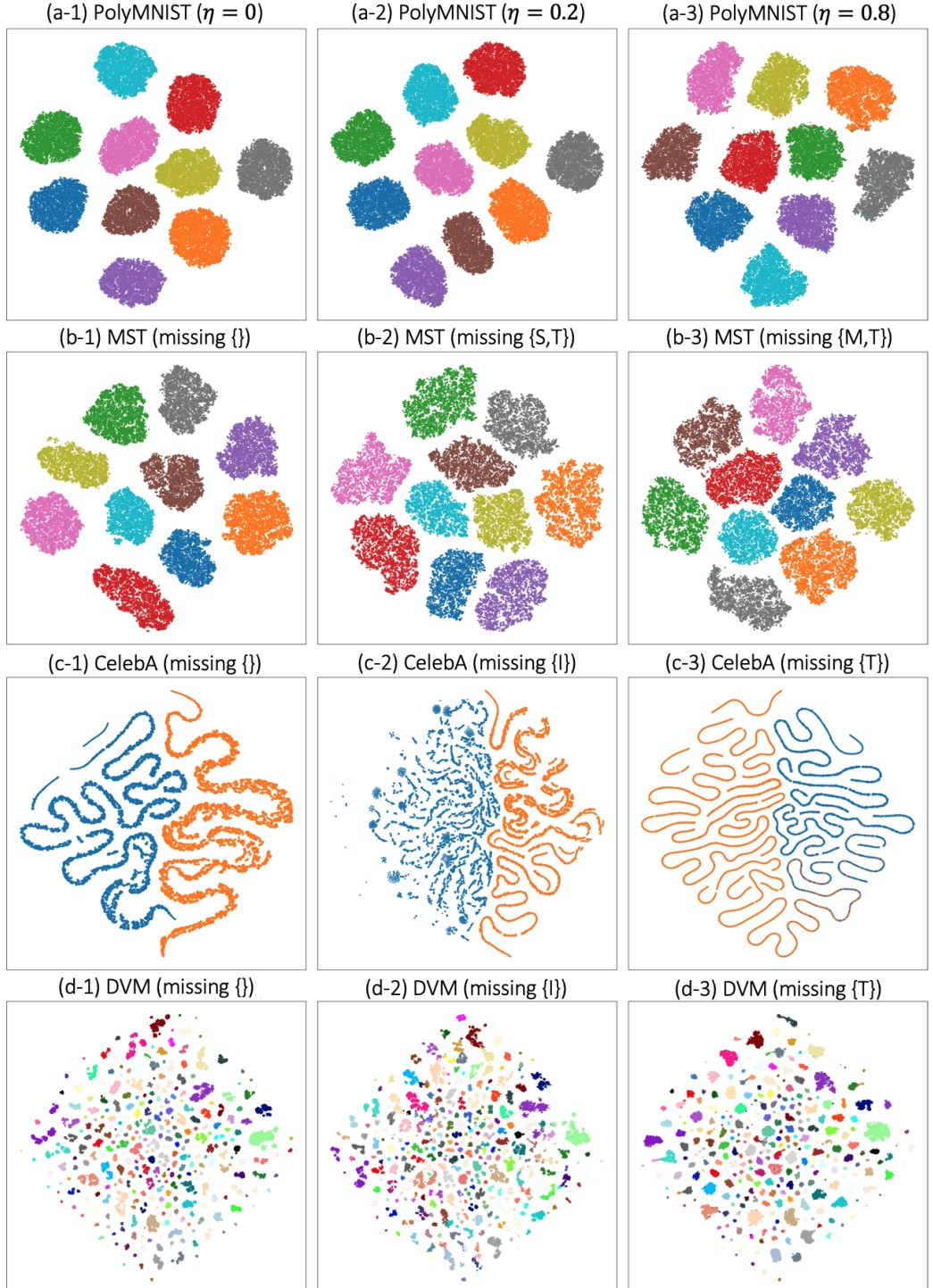

Figure S4: t-SNE visualization of DyMo$_c$'s multimodal network on the training data of (a) PolyM-NIST, (b) MST, (c) CelebA, and (d) DVM under various missing scenarios. Colors denote different class labels.

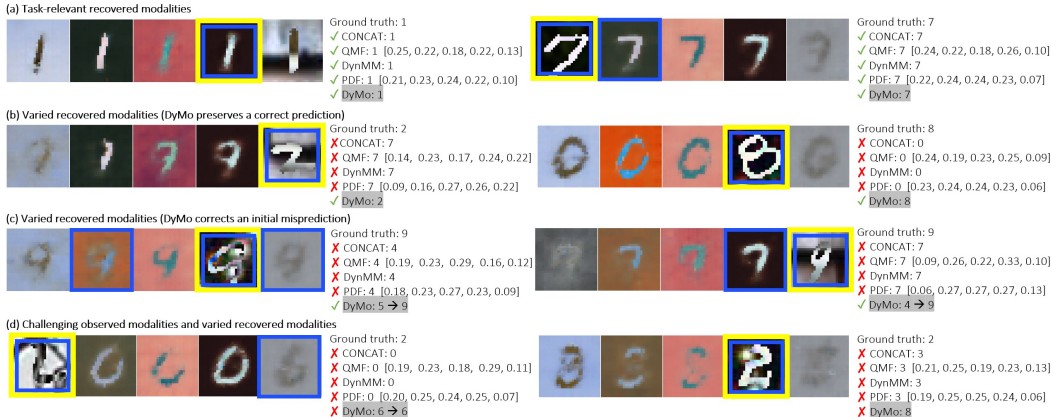

Figure S5: Extending the cases shown in Fig. 5(b), this figure presents additional representative examples of model predictions on PolyMNIST with missing rate $\eta = 0.8$ for DyMo and static/dynamic multimodal fusion methods. The sub-figures illustrate: (a) all recovered modalities are task-relevant; (b) recovered modalities vary in quality, and DyMo preserves correct predictions by disregarding unreliable recoveries; (c) recovered modalities vary in quality, and DyMo corrects initial mispredictions by incorporating task-relevant recovered modalities; (d) particularly challenging cases with limited task-relevant information in both non-missing and recovered modalities. Yellow boxes indicate non-missing modalities, and blue boxes indicate modalities selected by $DyMo_c$. $\checkmark$ denotes correct predictions, while $\times$ denotes incorrect predictions. For QMF and PDF, which perform dynamic weighted fusion, we report the weights assigned to each modality for every sample.

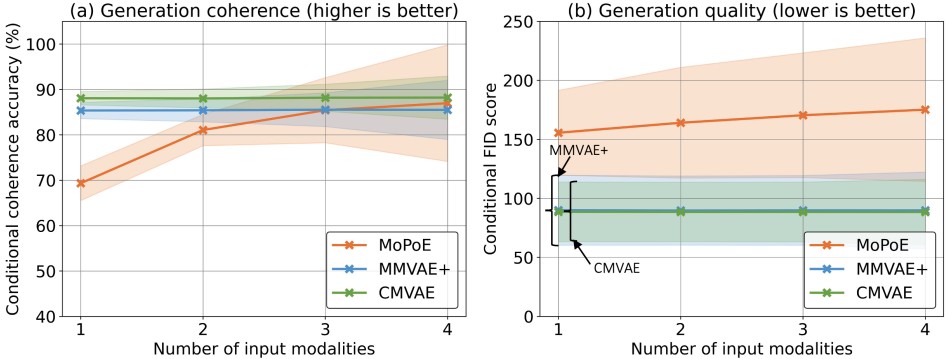

Figure S6: Reconstruction performance of different VAE-based recovery methods on PolyMNIST: (a) reconstruction coherence (higher is better), (b) reconstruction quality measured by FID (lower is better).

**Table Reconstruction Analysis:** Tab. S7 and Fig. S7 report the reconstruction results of tabular features for DVM and UKBB under full-table missingness, respectively. Categorical features are evaluated using accuracy, while continuous features are assessed with mean squared error (MSE). The results show that most tabular features are accurately reconstructed, *e.g.*, *color* in DVM (Acc: 82.46%), suggesting that their integration can benefit the target classification task. Some tabular features, however, exhibit low reconstruction performance, likely due to their relatively weak correlations with the image modality. Examples include *fuel type* in DVM (Acc: 64.45%) and *smoking status* in UKBB (Acc: 4.4%). In real-world applications, though, tabular features are more often partially missing rather than entirely absent (Wu et al., 2024b; Xue et al., 2024). Under a tabular missing rate $\gamma = 0.5$, we observed that the reconstruction performance of these challenging features improves substantially, *e.g.*, *fuel type* in DVM (64.45% → 87.13%) and *smoking status* in UKBB (4.4% → 86.99%). These findings suggest that the proposed DyMo can generalize well to practical scenarios, where partial missingness is more common than than full-table missingness.

Table S7: Reconstruction performance of 17 tabular features (4 categorical and 13 continuous) using TIP on DVM under full-table missingness (*i.e.*, $\gamma = 1$). **Cat** denotes whether a tabular feature is categorical, and $N_{unq}$ represents the number of unique values for each categorical feature.

| Tabular Feature | Cat | $N_{unq}$ | MSE ↓ | Acc (%) ↑ | Tabular Feature | Cat | $N_{unq}$ | MSE ↓ | Acc (%) ↑ |
|---|---|---|---|---|---|---|---|---|---|
| Advertisement month (Adv_month) | × | - | 1.2340 | - | Height | × | - | 0.3234 | - |
| Advertisement year (Adv_year) | × | - | 0.4448 | - | Length | × | - | 0.4272 | - |
| Bodytype | √ | 13 | - | 82.25 | Price | × | - | 0.2921 | - |
| Color | √ | 22 | - | 82.46 | Registration year (Reg_year) | × | - | 0.2622 | - |
| Number of doors (Door_num) | × | - | 0.3641 | - | Miles runned (Runned_Miles) | × | - | 0.6962 | - |
| Engine size (Engine_size) | × | - | 0.3818 | - | Number of seats (Seat_num) | × | - | 0.4601 | - |
| Entry prize (Entry_prize) | × | - | 0.2908 | - | Wheelbase | × | - | 0.5938 | - |
| Fuel type (Fuel_type) | √ | 12 | - | 64.45 | Width | × | - | 0.5873 | - |
| Gearbox | √ | 3 | - | 73.24 | | | | | |

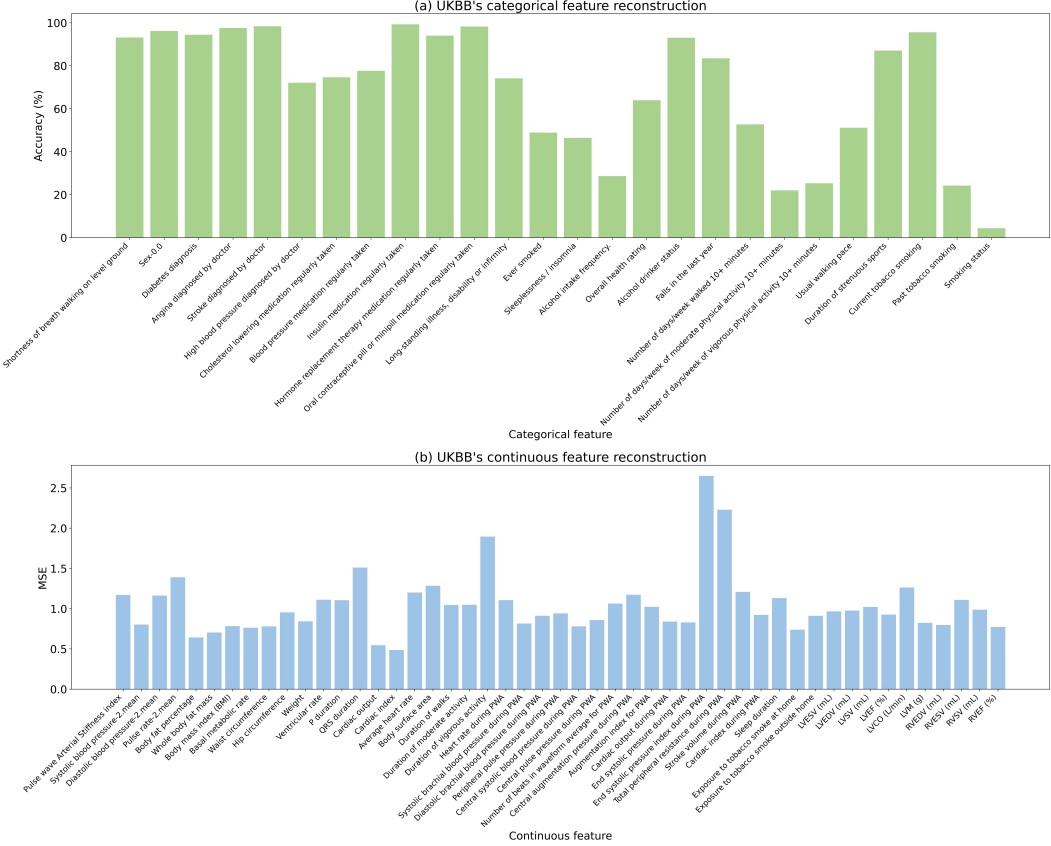

Figure S7: Reconstruction performance of 75 tabular features (26 categorical and 49 continuous) using TIP on UKBB under full-table missingness (*i.e.*, $\gamma = 1$).

# D DISCUSSION

**Beyond Classification Tasks:** While this paper primarily focuses on classification tasks, the DyMo's framework can be extended to a broader range of multimodal tasks. The key adaptation is to replace the cross-entropy (CE) term with the appropriate likelihood-based loss for the target task. Many multimodal tasks, *e.g.*, detection, segmentation, and sequence-to-sequence modelling, can be trained using probabilistic losses, and thus the same mutual information decomposition, $I(Y; Z) = H(Y) - H(Y|Z)$, remains applicable. In specific: (1) segmentation: MTIR can operate on the averaged per-pixel CE loss; (2) detection: MTIR can incorporate both classification and localization likelihoods as the task loss; and (3) sequence-to-sequence modelling: MTIR can aggregate token-level CE losses to guide modality selection.

