# OpenReview forum: "Inference-Time Dynamic Modality Selection for Incomplete Multimodal Classification"
_ICLR.cc/2026/Conference — ICLR 2026 Poster_

### Official Review · Reviewer_LF6X · 2025-10-28

**Soundness:** 3
**Presentation:** 3
**Contribution:** 3
**Rating:** 4
**Confidence:** 4

**Summary:**

**Summary**
To address the discard-fill dilemma faced by multimodal deep learning (MDL) in real-world scenarios due to modality loss， discarding the missing modality easily loses task-critical information, while filling the modality easily introduces low-fidelity/semantic misalignment noise. This paper proposes DyMo, a dynamic modality selection framework for inference, which aims to balance the utilization of missing modal information and noise avoidance.

**Strengths:**

**Strength**
1. The motivation is clear and convincing
2. The dataset and tasks are comprehensive. It covers five significantly different datasets, digital classification, attribute classification, disease diagnosis, and missing scenarios, fully verifying DyMo's adaptability in different scenarios.

**Weaknesses:**

**Weakness**
1. The primary concern is whether the noise in cross-modal generated data actually harms, especially with the advancement of diffusion model generation technology in recent years. The author should discuss and compare his work with the related recovery-based methods[1][2][3].

2. The comparison method is a bit outdated, and the author should consider comparing it with more 2024s methods of incomplete multimodal learning.

3. The experiments should contain some classical multimodal datasets, such as CMU-MOSI or CREMAD, for a fair comparison. especially the CMU-MOSI, in which the text is the dominant modality, when it misses the performance degree. This can help to verify the effectiveness of the DyMo.

4.The method requires the label information to select the modality, but when the modality is missing, the prediction will be very incorrect. How can the performance of the model be guaranteed? The representation prototypes also cannot avoid the problem of representations being misclassified.



[1] Yuanzhi Wang.  Incomplete Multimodality-Diffused Emotion Recognition NIPS
[2] S Wei. Mmanet: Margin-aware distillation and modality-aware regularization for incomplete multimodal learning CVPR
[3] Yuntao Shou. GSDNet: Revisiting Incomplete Multimodality-Diffusion Emotion Recognition from the Perspective of Graph Spectrum

**Questions:**

see the weakness

---

> ### Author Response · Authors · 2025-11-23
> **To Reviewer LF6X [Part I]**
>
> Thanks for your constructive comments and suggestions, they are greatly beneficial in enhancing the quality of our paper.  We have carefully incorporated them in the revised paper. Please see our responses below, which we hope will effectively address your concerns.
>
> > Q1: Discussion for noise in cross-modal generated data ------------------
>
> We clarify that the recovery noise we study (both low fidelity and semantically misaligned reconstructions) **does** harm downstream performance. This is consistently observed across all our benchmarks using the well-established recovery methods (MoPoE, MVAE++, CMVAE, TIP), all of which are common choices in their respective benchmarks. Our experiments show that DyMo largely outperforms the baseline that integrates all recovery without selection (+12.4% accuracy on PolyMNIST, Tab 2), and archives clear gain over static/dynamic fusion SOTAs (+4.11% accuracy on DVM, Fig. 3). These results empirically show the presence and negative impact of noise in cross-modal generated data.
>
> While diffusion models have improved low-fidelity reconstruction [1,3] (references suggested by the reviewer), recent studies show that diffusion models **still** suffer from semantic discrepancy issues [Ref2,Ref3], which closely relate to the semantically misalignment considered in our work. Prior work finds that controllable generation in diffusion models remains challenging [Ref1], especially when the conditioning signal provides limited task-relevant information, e.g., generating from partial observed modalities (CMVAE combined with diffusion models, Fig.15 (Palumbo et al., 2024)). As shown in IMDer [1] (reference suggested by the reviewer) , reconstructed samples can drift toward other-class distributions (their Fig. 3 and Fig. 4), further illustrating semantic issues.
>
> Another key limitation of diffusion-based reconstruction is computational cost. Conditional diffusion models typically require hundreds of denoising steps per modality (Palumbo et al., 2024), leading to significantly higher inference latency than VAE/AE-based methods. This becomes even worse when reconstructing multiple modalities (e.g., 5 modalities in PolyMNIST). Consequently, across 5 benchmarks used in our paper, the de facto SOTA recovery models used across our five benchmarks are predominantly VAE/AE-style models, which we adopt following common practice. To the best of our knowledge, diffusion-based works for image-table reconstruction are still very limited.
>
> It is important to note that the goal of our work is **not** to propose a new recovery model or to compete with diffusion-based models. Even if stronger recovery models emerge, the central question we investigate remains fundamental:
> *When recovery is imperfect, should all recovered modalities be trusted equally?*
> Our extensive experiments across 5 diverse datasets shows that the answer is no, and DyMo offers an effective mechanism for dynamic integration.
>
> IDMer [1] and GSDNet [3] (references suggested by the reviewer) reconstruct latent representations of missing modalities via diffusion models and jointly train with downstream emotion recognition tasks. These approaches fall under the online recovery-based methods (Sec. 2).  In contrast, DyMo operates on top of existing input-level reconstruction models and dynamically selects reliable recovered modalities. Our paper already covers representative online recovery-based methods (M3Care and OnlineMAE, method details in Appendix B.2), and we have now added IDMer and GSDNet to the related work (Sec. 2). We also note that GSDNet is from an IJCAI 2025 paper published only one month before the ICLR 2026 submission.
>
> MMANet [2] (reference suggested by the reviewer) trains a teacher model with complete modalities to guide a deployment model receiving incomplete data. Its missing-modality strategy uses zero-imputation, which falls under the recovery-agnostic category, a subset of recovery-free methods that focuses on learning modality-agnostic representations (Sec. 2). Our paper already includes representative methods in this category (ModDrop and MUSE, method details in Appendix B.2), and we have added MMANet to the related work as well (Sec. 2).
>
> Overall, IDMer, GSDNet, and MMANet belong to recovery-based or recovery-free methods, without considering the discarding-imputation dilemma (Sec. 2). In contrast, DyMo introduces a new perspective by dynamically selecting reliable recovered modalities, going beyond the traditional binary choice.
>
> [Ref1] Cao, Hanqun, et al. "A survey on generative diffusion models." IEEE transactions on knowledge and data engineering 2024.
>
> [Ref2] Zheng, Ziyang, Ruiyuan Gao, and Qiang Xu. "Non-Cross Diffusion for Semantic Consistency." 2025 IEEE/CVF Winter Conference on Applications of Computer Vision (WACV). IEEE, 2025.
>
> [Ref3] Liu, Qihao, et al. "Intriguing properties of text-guided diffusion models." CVPR 2023.

---

> > ### Author Response · Authors · 2025-11-23
> > **To Reviewer LF6X [Part II]**
> >
> > > Q2: Comparison methods ------------------
> >
> > We have already included recent SOTA models across all major categories of both static/dynamic multimodal fusion and incomplete multimodal learning (recover-based and recovery-free). Specifically, our comparisons cover:
> >
> > - Static/Dynamic fusion methods: static (CONCAT), dynamic (PDF, DynMM, QMF)
> > - Recovery-based incomplete methods: offline (MultiAE, MoPoE) and online (M3Care and OnlineMAE)
> > - Recovery-free incomplete methods: modality-agnostic (ModDrop and MUSE) and modality-aware (MTL and MAP).
> >
> > Notably, many of these methods, MUSE (ICLR’24) , PDF (ICML’24), QMF (ICML’23), MAP (CVPR’23), OnlineMAE (AAAI’23), DynMM (CVPRW’23), MAP (CVPR’22), M3Care (KDD’22), and MoPoE (ICLR’21), are pretty much recent and representative SOTAs within their respective category.
> >
> > To further address the reviewer's suggestion, we additionally evaluate CMVAE (Palumbo et al., ICLR 2024) on PolyMNIST under different modality missing rates $\eta$. Because CMVAE does not simulate missing modalities during training, we also incorporate our incomplete simulation training to obtain CMVAE$^{\dagger}$).
> >
> > | Model| Year | $\eta$=0.0 | 0.2 | 0.4 | 0.6 | 0.8 |
> > |-------|-----|-----|-----|-----|-----|-----|
> > | PDF | 2024 | 99.99 | 99.93 | 99.37 | 95.84 | 82.71 |
> > | MUSE | 2024 | 99.93 | 99.64 | 98.61 | 94.73 | 77.56 |
> > | CMVAE (newly added) | 2024 | 94.93 | 95.50 | 95.21 | 95.13 | 95.11 |
> > | CMVAE$^{\dagger}$ (newly added) | 2024 | 94.59 | 95.23 | 94.93 | 95.17 | 95.20|
> > | DyMo$_e$ | - | **100** | **100** | **99.99** | **99.87** | **96.81** |
> >
> > CMVAE remains relatively stable across missing rates, mainly because it performs classification using a single randomly selected observed modality. This avoids interference from missing modalities but limits its ability to exploit information from additional available ones. In contrast, DyMo consistently outperforms all 2024 SOTAs, demonstrating the effectiveness of our dynamic modality fusion across various missing rates.
> >
> > We have now included this experiment in the revised paper (Tab. S3).
> >
> > > Q3: New multimodal dataset ------------------
> >
> > We highlight that our evaluation already spans a diverse set of well-established multimodal benchmarks across natural and medical domains, covering multiple modality types (image, text, and tabular data). PolyMNIST, MST, and CelebA are long-standing benchmarks in multimodal learning and missing-modality research (Sutter et al., ICLR 2021; [Ref4] ICLR 2024; Gao & Pu, ICLR 2025),  while DVM and UKBB are widely used benchmarks in multimodal image-tabular medical analysis (Hager et al., CVPR 2024; Du et al., ECCV 2024; Du et al., CVPR 2025). These datasets are broadly adopted in recent SOTA studies and are considered representative testbeds for evaluating multimodal fusion under varying missing modality conditions.
> >
> > We clarify that our work primarily focuses on multimodal tasks with static modalities (image, text, table). In contrast, CMU-MOSI and CREMAD are emotion recognition datasets that involve temporal modalities (video and audio), which introduce sequence modeling considerations beyond the scope of this study. Our framework can be extended to temporal settings by using standard temporal encoders as modality-specific encoders, but this is orthogonal to our main contributions and is left for future work (as mentioned in the conclusion, Line 491-492).
> >
> > Regarding the reviewer’s concern about dominant text modalities, as exemplified by CMU-MOSI. We highlight that this scenario has already been included and thoroughly examined in our experiments. CelebA, MST, and CMU-MOSI share a common characteristic: their text modality contains clean, high-level semantic information that directly aligns with the target prediction, making it inherently more discriminative than visual or audio modalities that often contain noisier or less direct signals.
> > - CelebA: attribute descriptions (e.g., “blonde hair”) explicitly reveal the target attributes.
> > - MST, textual labels map exactly the ground-truth digit classes.
> > - In CMU-MOSI, transcript text contains explicit sentiment-bearing cues.
> > This structure naturally leads to text-dominant behavior, and the validation scenario highlighted by CMU-MOSI is already reflected in CelebA and MST.
> >
> > Our results show that missing text indeed substantially degrades the performance of prior methods (Tab. 1). Recovery-free methods suffer even more than recovery-based methods, because they directly ignore missing yet highly task-relevant text and rely only on less discriminative available modalities (Fig. 1(a), Tab. 1, Line 47-53, Line 418-421). Prior recovery-based methods also struggle due to varying reconstruction qualities across samples (Tab. 1, Fig. 1 (b)). These trends reflect the discard-or-impute dilemma that motivates DyMo.

---

> > > ### Author Response · Authors · 2025-11-23
> > > **To Reviewer LF6X [Part III]**
> > >
> > > By selectively integrating reliable recovered modalities, DyMo effectively mitigates the impact of missing dominant text. On both CelebA and MST, DyMo significantly outperforms prior recovery-based and recovery-free approaches (e.g, +3.88% accuracy on CelebA, Tab. 1).
> > >
> > > [Ref4] Cui, Sen, et al. "CLAP: Collaborative adaptation for patchwork learning." ICLR 2024.
> > >
> > > > Q4: Label information concern ------------------
> > >
> > > DyMo already accounts for incorrect label predictions made using only the observed modalities. Such errors is an inherent limitation of recovery-free methods and reflects one side of the *discarding-imputation dilemma* (Fig.1 (a)). Prototype-based prediction cannot fully eliminate misclassification under missing modalities. However, DyMo is built upon an explicit and well-motivated assumption (detailed below) about when such misclassification occurs and how recovered modalities can help mitigate it.
> > >
> > > Misclassification with missing modalities typically occurs when the available modalities contain insufficient task-relevant information. In such cases, the resulting representation often lies near class boundaries and may be relatively far from the true class prototype. This phenomenon has been reported in prior studies [Ref6,Ref7] and also reflected in our Fig. 1(a) and Fig. 4 (a). DyMo’s MTIR reward is designed around two principles:
> > >
> > > (1) Informative recovered modalities tend to shift the representation toward a more discriminative region and closer to the appropriate class prototype
> > > (2) Unreliable recovered modalities may cause inconsistent shifts, which DyMo learns to identify and disregard.
> > >
> > > To further enhance robustness, we introduce a calibration term that measures how representative a sample is within its predicted class cluster (Sec. 3.2). This encourages DyMo to select only the recovered modalities that move the multimodal representation closer to the predicted class prototype and increase its representativeness within the cluster.
> > >
> > > Our visualizations (Fig. 4, Fig. 5, Fig. S4) show that, when the recovered information is sufficiently reliable, DyMo can correct initially incorrect predictions by incorporating these modalities, and can maintain correct predictions by disregarding unreliable recovery. In addition, our experiments show that DyMo significantly outperforms prior recovery-based and recovery-free SOTAs (Tab. 1), as well as a baseline that integrates all modalities without selection (Tab. 2). This suggests that either blindly discarding or integrating the recovered modalities is limited in incomplete multimodal learning, and that dynamic modality selection offers a more effective alternative..
> > >
> > > We also show particularly challenging cases for all models (4th row in Fig. 5, Fig. S4(c)) , where both non-missing and recovered modalities provide limited task-relevant information. In such cases, DyMo may not fully correct initial errors, suggesting that more advanced recovery methods could further enhance performance.
> > >
> > > Finally, using model predictions from a subset of modalities as proxy labels is a common and accepted practice when ground-truth labels are unavailable (Zhang et al., 2023; Cao et al., 2024; [Ref5]). Our outperformance over SOTAs empirically show the design is appropriate for our setting.
> > >
> > > [Ref5] Wang, Xiaoli, et al. "Trusted semi-supervised multi-view classification with contrastive learning." IEEE Transactions on Multimedia 2024.
> > >
> > > [Ref6] Van Engelen, Jesper E., and Holger H. Hoos. "A survey on semi-supervised learning." Machine learning 2020.
> > >
> > > [Ref7] Karimi, Hamid, Tyler Derr, and Jiliang Tang. "Characterizing the decision boundary of deep neural networks." arXiv 2019.

---

> > > > ### Comment · Reviewer_LF6X · 2025-11-25
> > > >
> > > > The authors have addressed my concerns, and I will raise my score.

---

> > > > > ### Author Response · Authors · 2025-11-26
> > > > > **To Reviewer LF6X [Thank you]**
> > > > >
> > > > > Thank you for taking the time to review our rebuttal. We sincerely appreciate your engagement and are glad that our response has addressed your concerns. We are grateful for your constructive feedback and for reconsidering your evaluation.

---

### Official Review · Reviewer_eaDt · 2025-10-28

**Soundness:** 3
**Presentation:** 2
**Contribution:** 3
**Rating:** 6
**Confidence:** 4

**Summary:**

This paper proposes a Dynamic Modality selection method, DyMo, for the missing modality machine learning setting. The key insight is that existing approaches either discard missing modalities (losing valuable information) or recover all missing modalities (potentially introducing noise), creating a fundamental trade-off the authors refer to the discarding-impuattion dilemma. DyMo overcomes this by adaptively selecting only the reliable recovered modalities that provide task-relevant information for each test sample.​

- Core Innovation: A Multimodal Task-Relevant Information Reward (MTIR) function that estimates incremental information gain from each recovered modality using task loss as a tractable proxy​

- Theoretical connection: MTIR is based in a connection between mutual information I(Y;Z) and classification loss, enabling inference-time selection without ground-truth labels​

- Evaluation: Extensive experiments on 5 datasets (PolyMNIST, MST, CelebA, DVM, UKBB) showing significant improvements, especially under severe missing scenarios​

**Strengths:**

- Strong Problem Formulation: This work substantiates discarding-imputation dilemma in incomplete multimodal learning providing a strong motivation for DyMo

- Comprehensive Technical Design: MTIR reward function handles both low-fidelity and semantically misaligned recovered modalities​ (e.g. when the image is blurry or the recovered image is not representative of the class)

- Strong additional features: Intra-class similarity calibration enhances reward reliability​, iterative modality selection to minimize noise, incomplete simulation training, auxiliary contrastive loss tested with 2 distance functions

- Extensive Experimental Validation: Evaluation across diverse domains (natural images, medical data, synthetic benchmarks)​, Consistent improvements over 12 baseline methods​, particularly strong performance under severe missing scenarios (e.g., 13.12% improvement on PolyMNIST with 80% missing modalities)​

- Thorough Analysis: Comprehensive ablation studies and visualization analyses (Figure 4 with the TSNE and PCA visualizations were particularly convincing that DyMo’s MNITR successfully adds recovered modality features when helpful and does not use them when it would hurt performance.

**Weaknesses:**

- Limited Recovery Method Diversity: While claiming generalizability, experiments primarily use VAE-based recovery methods (MoPoE, MMVAE+, CMVAE from Table S5) with limited evaluation of fundamentally different recovery approaches​

- Computational Overhead: The authors claim DyMo introduces minimal additional parameters and relies on a relatively simple training scheme. However, it seems give the features of the method including computing the MITR, which includes intra-class similarity calibration, and with iterative selection (average 1.38 iterations per sample from  appendix C.3), DyMo would be more computationally intensive. The inference-time latency, parameter count, or training time computational cost were not thoroughly analyzed​ in this work.

- Calibration Term Limitations: The intra-class similarity calibration shows inconsistent benefits across datasets (improves some tasks but hurts CAD/Infarction performance), suggesting the approach may not be universally optimal​

- Limited Analysis of Edge Cases and limitations: Insufficient discussion of common failure modes or limitations of when DyMo performs poorly or what assumptions DyMo makes.

**Questions:**

- Recovery Method Dependencies: How sensitive is DyMo's performance to the quality of the underlying recovery method? Could you provide analysis on performance degradation when recovery methods produce consistently low-quality reconstructions? For example, if the recovery method always produces noise how would DyMo perform. If it produces accurate recoveries 50% of the time, how well would DyMo perform?

- Computational Scalability Concerns: How does the computational overhead scale with the number of modalities and missing patterns? What is the practical upper limit for real-time applications?

- Hyperparameter Sensitivity: The framework introduces several hyperparameters (temperature t=0.1, calibration threshold alhpa). How sensitive is performance to these choices, and how should they be set for new domains?

- Theoretical Limitations: The mutual information lower bound assumes bounded loss values with conservative upper bound G. How is G estimated in practice, and how does this choice affect the bound's tightness?

- Class Imbalance: How does DyMo perform on highly imbalanced datasets where the equal class prior assumption may be violated? Could you provide analysis or modifications for such scenarios?

- Generalization Beyond Classification: While focused on classification, could this approach be extended to other multimodal tasks like regression, generation, or structured prediction? What modifications would be required?

---

> ### Author Response · Authors · 2025-11-23
> **To Reviewer eaDt [Part I]**
>
> We thank the reviewers' positive and valuable feedback, which has greatly helped improve our paper. We have carefully incorporated them in the revised paper. Please see our responses below, which we hope will effectively address your concerns.
>
> > W1: Recovery method diversity ------------------
>
> We clarify that DyMo is **not** restricted to VAE-based recovery methods. In the DVM and UKBB experiments, we use TIP, a masked-autoencoder (MAE)-based recovery model, which is fundamentally different from VAEs.
>
> Our goal is **not** to exhaustively evaluate every possible recovery module, but to evaluate DyMo with the commonly used or SOTA recovery methods specific for each dataset. The main contribution is to introduce a new dynamic modality selection perspective for addressing the *discarding-imputation dilemma* in incomplete multimodal learning. Therefore, PolyMNIST/MST/CelebA use MoPoE (ICLR'21), MMVAE+ (ICLR'23), and CMVAE (ICLR'24), and DVM/UKBB use TIP (ECCV'24).
>
> To directly address the reviewer's concern about recovery diversity, we further include a machine-learning-based recovery method, iterative multivariate imputer (IMI), a widely-used table imputation method that iteratively recovers missing values using correlations among table columns [Ref1].
>
> | Model | DVM 0.7 | DVM 0.9 | UKBB 0.7 | UKBB 0.9 |
> |-------|-----|-----|-----|-----|
> | CONCAT (IMI) | 81.85 | 67.13 | 73.75 | 69.39 |
> | DyMo$_e$ (IMI)  | **95.97** | **94.28** | **75.96** | **71.09** |
>
> These results show that DyMo continues to provide strong improvements even when paired with a non–deep learning, multivariate imputation method, supporting the robustness of the framework across different recovery families. We have added the IMI experiments to Appendix C.2 and Fig. S6.
>
> [Ref1] Liu, Jingchen, et al. "On the stationary distribution of iterative imputations." Biometrika 2014.
>
> > W2 & Q2: Computational overhead & computational scalability ------------------
>
> We clarify that our statement “DyMo introduces minimal additional parameters and relies on a relatively simple training scheme” (Line 356-358, original paper) is **not** intended to claim low inference-time cost. Instead, it highlights DyMo’s simplicity relative to prior dynamic fusion methods (DynMM, PDF, QMF). Specifically, PDF and QMF require additional modality-specific branches for importance estimation, and DynMM involves two-stage training (training unimodal/multimodal branches followed by a gating network). In contrast, DyMo uses a single-stage training pipeline, and its dynamic algorithm operates directly on multimodal representations without introducing extra parameters. We have revised the sentence to *“Notably, prior dynamic methods typically require additional modality-specific branches for modality contribution estimation or multi-stage training, whereas DyMo operates directly on multimodal representations without introducing extra modality-specific parameters and relies on single-stage training.”* (Line 363-366) to avoid ambiguity.
>
> To directly address the reviewer's concern about computation overhead, we report parameter counts, training time, and inference latency on PolyMNIST (the dataset with the largest number of modalities) when missing 80% modalities:
>
> | Model | Train/Inference param. (M) | Train time (min/epoch) | Inference latency (ms/sample) | Acc (%) |Accuracy gap to DyMo Performance gap (%) |
> |-------|-----|-----|-----|-----|-----|
> | CONCAT | 5.76/5.76 | 0.17 |  0.0035 | 83.69 | -13.72|
> | PDF | 10.99/10.99 | 0.16 | 0.0062 | 82.71 | -14.16 |
> | M3Care | 2.59/2.59 | 0.64 | 0.0168 | 87.92 | -8.89 |
> | MTL | 0.95/0.95 | 1.81 | 0.0169 | 91.14 | -5.67 |
> | DyMo | 0.96/0.96 | 0.35 | 0.0469 | **96.82** | 0 |
>
> These results indicate that while DyMo has slightly higher inference latency, it maintains a comparable parameter count and training cost, and provides a favorable tradeoff between efficiency and predictive performance.
>
> For computational scalability concern, DyMo’s overhead scales linearly with the number of missing modalities. If the MITR cost per modality is T, and a sample has C missing modalities, each iteration adds ≈ T × C latency. Since DyMo converges in 1–2 iterations on average, the overall overhead remains modest. Importantly, the cost depends on C rather than the missing patterns. In our experiments with 5 modalities and an 80% randomly missing rate, DyMo’s latency is 0.0469 ms/sample, and real-time use remains feasible when only a few modalities are missing.

---

> > ### Author Response · Authors · 2025-11-23
> > **To Reviewer eaDt [Part II]**
> >
> > > W3: Calibration term limitation ------------------
> >
> > We discussed this issue in the paper (Line 447-451). The intra-class similarity calibration term can reduce performance on some datasets, likely because it makes DyMo more conservative in modality selection. To validate this explanation, we introduced a simple scaling hyper-parameter $\epsilon$ applied to $\alpha$, i.e., $\alpha' = \alpha \times \epsilon$, and evaluated on CAD with missing {T}.
> >
> > |$\epsilon=$| 1 | 2 | w/o calibration |
> > |------|------|------|------|
> > |AUC | 71.20 | 71.84 | 72.20 |
> >
> > The results show that the performance drop can be mitigated by adjusting a single scaling hyperparameter, suggesting that the degradation is likely due to suboptimal scaling for this specific dataset rather than a fundamental limitation of the calibration design.
> >
> > Finally, we emphasize that our intra-class similarity calibration is simple yet effective overall:
> > - it avoids expensive non-parametric density estimates requiring all training samples (Line 251-252) [Ref2]
> > - it consistently improves performance on most datasets(e.g., +3.19% accuracy on MST and +8.53% on CelebA), although tuning the scaling factor may be helpful for more complex datasets.
> >
> > [Ref2] Hastie, Trevor, Robert Tibshirani, and Jerome Friedman. "The elements of statistical learning." (2009).
> >
> > > W4: Failure case discussion ------------------
> >
> > We have already shown and discussed failure cases for DyMo in Sec. 4.2 (the 4th row in Fig. 5) and Appendix C.4 (Fig. S5(d)). As stated in the paper:
> >
> >  *“Fig.4(d) illustrates particularly challenging cases for all models, where both non-missing and recovered modalities provide limited task-relevant information. In these rare and difficult cases, DyMo may not fully correct its initial mispredictions, suggesting that the use of more advanced recovery methods could further enhance performance.”*
> >
> > These examples explicitly highlight DyMo’s limitations and potential solutions.
> >
> > > Q1: Recovery method dependency ------------------
> >
> > We argue that DyMo is robust to the reconstruction quality. Our results show that DyMo remains stable across different recovery methods, e.g., CMVAE, MVAE++, and MoPoE, despite their varying reconstruction performance (Tab. S5 and Fig. S5). This suggests that DyMo’s selection performance is **not** highly sensitive to the quality of recovered modalities.
> >
> > To directly address the reviewer's concern about DyMo's sensitivity to recovery quality, we additionally conduct an extreme simulation experiment with a controlled correct recovery rate $r$. For $r \times 100%$ of the samples, the missing modalities are imputed with their ground-truth versions; for the remaining $(1-r)\times 100%$, the missing modalities are replaced with zero noise. We evaluate DyMo on MST (missing {M,T}) and CelebA (missing {T}).
> >
> > |Correct recovery rate $r$ | MST 0.7 | MST 0.5 | MST 0.0 | CelebA 0.7 | CelebA 0.5 | CelebA 0.0 |
> > |-------|-----|-----|-----|-----|-----|-----|
> > | CONCAT | 72.99 | 55.05 | 10.31 | 95.81 | 93.12 | 86.67 |
> > | PDF | 83.36 | 72.38 | 44.89 | 95.61 | 93.10 | 86.67 |
> > | DyMo$_c$ | **89.00** | **86.04** | **78.29** | **97.04** | **96.51** | **95.14** |
> >
> > The results demonstrate that prior static and dynamic fusion methods degrade sharply as recovery quality deteriorates. In contrast, DyMo remains relatively stable across a wide range of correct recovery rates, suggesting its ability to effectively disregard unreliable recovered modalities.
> >
> > We have added the details of this simulation and the corresponding analysis to Appendix C.2 and Fig. S2.

---

> > > ### Author Response · Authors · 2025-11-23
> > > **To Reviewer eaDt [Part III]**
> > >
> > > > Q3: Hyperparameter sensitivity ------------------
> > >
> > > We clarify that the calibration term $\alpha$ is **not** a user-defined threshold hyperparameter. Instead, it is a scale parameter computed by DyMo based on sample representations via Eq. 8 and Eq. 9. Our ablation studies (Tab. 2) and visualizations (Fig. 4(b)) show the effectiveness of this learnt calibration mechanism. For the temperature parameter, we follow common practice in metric learning (Hager et al., 2023; Du et al., 2024; [Ref3]) and set $t=0.1$ for all datasets.
> > >
> > > To directly address the reviewer's concern regarding temperature parameter sensitivity, we evaluate DyMo under different temperature values on MST with missing {T}.
> > >
> > > | Temperature $t$ | 0.05 | 0.1 | 0.15 | 0.2 | 0.5 | 1.0 |
> > > |-------|-----|-----|-----|-----|-----|-----|
> > > | Acc (%) | 98.96 | 98.99 | 99.10 | 98.96 | 99.08 | 99.07 |
> > >
> > > The results show that DyMo remains stable across a range of temperatures, indicating that the model is relatively insensitive to the choice of $t$.
> > >
> > > For new domains, since DyMo is relatively insensitive to $t$, a small fixed value such as $t=0.1$ can be used as a default without tuning. If tuning is desired, practitioners may perform a coarse sweep (e.g., ${0.05, 0.1, 0.2}$), which requires little effort and consistently yields stable results in our experiments.
> > >
> > > Finally, we highlight that this paper already included sensitivity analysis and discussion of the key hyper-parameters introduced by DyMo, such as the number of sampled modality subsets ($A$) used in the incomplete-modality simulation training (Appendix C.2, Fig. S1 (a)).
> > >
> > > [Ref3] Chen, Ting, et al. "A simple framework for contrastive learning of visual representations." ICML 2020.
> > >
> > >
> > > > Q4: Theoretical concern ------------------
> > >
> > > Our MI-CE lower bound is formulated as (Eq. 2 and Eq. S9):
> > >
> > > $$
> > > I(Y,Z) \geq H(Y) - \hat{L}_{ce} - G \sqrt{\ln{(1/\delta)}/2|D|}
> > > $$
> > >
> > > In this bound, the dominant term is the CE loss. Both H(Y) and the G-related term are constants w.r.t. DyMo’s dynamic selection process.
> > >
> > > We clarify that DyMo does not require explicitly estimating G in practice. In the theoretical derivation, G serves only as a conservative upper bound of the loss, and the bound’s third term is scaled by $1/\sqrt{|D|}$. For datasets of moderate size, this term becomes numerically negligible.
> > >
> > > To illustrate this, we report the empirical CE loss range and the resulting value of the G-related term on PolyMNIST with 60% missing modalities (taking G as the maximum CE loss and $\delta=0.1$):
> > >
> > > |Dataset| Size | Min | Max (G)|  G-related term in the MI-CE lower bound |
> > > |------|------|------|-----|-----|
> > > |PolyMNIST| 7,000 | 0 |  4.14 | 0.005349 |
> > >
> > > The contribution of the G-related term is extremely small (<0.006), and thus does not affect the lower bound tightness in practice. This also implies that G does not play an important role in DyMo’s dynamic selection, which only relies on changes in CE loss (Sec. 3.2). Therefore, no estimation or tuning of G is required in practice.
> > >
> > >
> > > > Q5: Class imbalance ------------------
> > >
> > > We clarify that DyMo does **not** rely on an equal class prior and one of our datasets (DVM) is in fact highly imbalanced.
> > >
> > > The equal prior assumption is used only in the derivation of prototype-based prediction (Eq. 5, Appendix A.2). Specifically, assuming a uniform prior allows the softmax over distances, $\exp(-d_{\phi}(z, c_k))$, to directly correspond to the likelihood term in the MTIR derivation without introducing additional class-dependent constants. This matches the standard formulations in prototype-learning (Snell et al., 2017) simplifies the MI–based analysis. Importantly, this assumption is not used in training DyMo nor does it require the dataset to be balanced.
> > >
> > > In practice, DyMo only requires that the same distance metric be used across all classes so that distances to prototypes are comparable. The dynamic recovery mechanism depends solely on these distances and remains unaffected by skewed class frequencies. As evidence, DyMo performs robustly on a highly imbalanced dataset (DVM): among 286 classes, sizes range from 1,980 to 40 samples (2% of the largest). DyMo achieves strong accuracy on both minority and majority classes (e.g., 52-sample class: 0.92; 1,323-sample class: 0.94), demonstrating stability under severe imbalance.
> > >
> > > If class prior estimates $\pi_k$ are available, one can replace the uniform prior with $p(y=k|z)=\pi_{k}\exp(-d_{\phi}(z,c_k))/\sum^K_{k'}\pi_{k'}\exp(-d_{\phi}(z,c_{k'}))$ (Eq. 5), and DyMo can incorporate this directly into the MTIR reward (Eq. 4). This only adds a class-dependent constant and does not require architectural changes.
> > >
> > > Finally, since DyMo's dynamic mechanism operates entirely at inference time, it remains compatible with common imbalance mitigation techniques, such as class re-weighting and re-sampling, if the practitioners wish to further improve minority-class performance.

---

> > > > ### Author Response · Authors · 2025-11-23
> > > > **To Reviewer eaDt [Part IV]**
> > > >
> > > > > Q6: Generalization beyond classification ------------------
> > > >
> > > > MTIR is presented in the classification setting because (1) CE-based classification is the common setting for information-theoretic analysis of representations (Tishby et al., 2015; Du et al., 2025; [Ref1]), and (2) it allows a clean derivation of the MI-CE lower bound that DyMo builds on.
> > > >
> > > > Extending DyMo to other multimodal tasks is generally feasible, as many tasks (e.g., segmentation/detection/seq-to-seq) also rely on likelihood-based objectives. Since MTIR requires an observable likelihood to approximate $H(Y|Z)$, the framework can naturally generalize by replacing the CE term with the corresponding task-specific likelihood. In specific: (1) segmentation: MTIR can operate on the averaged per-pixel CE loss; (2) detection: MTIR can incorporate both classification and localization likelihoods as the task loss; and (3) sequence-to-sequence modelling: MTIR can aggregate token-level CE losses to guide modality selection.
> > > >
> > > > As noted in the conclusion, exploring these extensions is part of our future work (Line 490-491). We now include a brief discussion in Appendix D the modifications required for applying DyMo to other multimodal settings.

---

### Official Review · Reviewer_V4dA · 2025-11-01

**Soundness:** 4
**Presentation:** 3
**Contribution:** 3
**Rating:** 6
**Confidence:** 3

**Summary:**

This paper tackles incomplete multimodal classification by explicitly framing the discarding–imputation dilemma and proposing DyMo, an inference‑time dynamic modality selection framework. The method recovers missing modalities via any recovery model, then iteratively selects only those recoveries that yield positive task‑relevant information according to a reward grounded in a connection between mutual information and cross entropy, refined with an intra‑class similarity (ICS) calibration, and implemented through a lightweight greedy procedure. The paper pairs this with a transformer‑based multimodal backbone that accepts arbitrary modality subsets and a simple incomplete‑modality simulation training scheme. Extensive experiments on five datasets (natural and medical) show consistent gains, especially at high missing rates.

**Strengths:**

- This paper perceptively identifies discarding–imputation dilemma in incomplete multimodal learning, which is an interesting and practical research problem.

- Selection criterion grounded in standard MI-CE relations and rendered computable via prototype‑based energies (Eq. 5–7), yielding an interpretable "move‑toward‑the‑prototype" test.

- Broad empirical coverage with competitive results on five datasets, strong robustness under severe missingness, and ablations demonstrating the contribution of each component.

- Good practicality: recovery‑method agnostic with positive results across MoPoE/MMVAE+/CMVAE and modest extra inference steps on average.

**Weaknesses:**

- Prototype posterior in Eq. 5–6 presumes Bregman divergences; the cosine distance used in $\text{DyMo}_c$ lacks that guarantee, and the authors claim that both $\text{DyMo}_c$ and $\text{DyMo}_e$ achieved similar results, indicating that DyMo is robust to the choice of distance metric. However, as shown in the results of Table 1, the performance of  $\text{DyMo}_c$ and $\text{DyMo}_e$ across different settings (datasets/missing rates/metrics) does not seem to be consistent.

- Sub-optimality of the greedy strategy: The algorithm relies on a greedy approach, selecting at each step the single modality that currently yields the highest reward. This strategy cannot guarantee finding the optimal combination of all missing modalities. There can be scenarios in which adding modality A alone is less beneficial than adding modality B, yet adding the combination of modalities A and C yields far greater gains than adding B alone. The effect of this kind of modality synergy can be significant in different medical imaging modalities. A greedy selection would overlook such combinatorial effects, leading to a sub-optimal subset of modalities.

- The author acknowledges that TIP has an upper limit for full-table reconstruction and uses this to explain why it performs on par with or worse than CONCAT in certain scenarios. However, this means that the "selection strategy" capability becomes strongly constrained by the reconstruction quality bottleneck. I highly recommend to include at least one alternative reconstructor in DVM/UKBB.

**Questions:**

- Could the authors provide a more in-depth analysis of the choice of distance metric, either theoretical or empirical?

- Please justify the selection of the greedy approach.

- Would the selection mechanism remain effective independent of the chosen reconstructor?

---

> ### Author Response · Authors · 2025-11-23
> **To Reviewer V4dA [Part I]**
>
> We sincerely appreciate your positive and valuable feedback, they are greatly beneficial in enhancing the quality of our paper. We have carefully incorporated them in the revised paper.  Please see our responses below, which we hope will effectively address your concerns.
>
> > W1 & Q1: Choice of distance metric  ------------------
>
> Our motivation for using different distance metrics is to evaluate how DyMo behaves under commonly used metrics in metric learning. The squared Euclidean metric is a Bergman divergence and therefore aligns with the theoretical assumptions in Eqs. 5-6. While the cosine distance does not strictly belong to the Bergman divergences, it is widely used in high-dimensional embedding spaces, where directional similarity is often more informative than vector magnitude (Line 880-885). We therefore include the cosine metric to empirically examine DyMo’s robustness when this theoretical assumption is relaxed.
>
> Our results show that DyMo with both metrics outperform prior incomplete/dynamic SOTAs on most datasets, especially under severe missingness (Fig. 3, Tab. 1). DyMo$_e$ (Euclidean) achieves similar or better performance than DyMo$_c$ (Cosine) in many cases, which is aligned with its stronger theoretical grounding. However, the performance difference is generally small, indicating that DyMo is not overly sensitive to the exact choice of metric and is robust to the exact choice of distance metric.
>
> Regarding our original claim "DyMo$_c$ and DyMo$_e$ yield similar results" (Line 363 in the original paper), our intention was to convey that both variants consistently outperform prior SOTAs across most datasets. We have revised this sentence to 'DyMo$_c$ and DyMo$_e$ consistently outperforms prior SOTAs on most datasets' (Line 370 in the revised paper) for improved clarity.
>
> > W2 & Q2: Greedy strategy for modality selection  ------------------
>
> Thank you for the thoughtful comment. We adopt an iterative strategy that selects the modality with the highest MTIR reward at each step based on the following considerations:
>
> (1) Reliability. Unlike common dynamic modality fusion settings (Zhang et al., 2023; Ma et al., 2019), our scenario is more challenging because it involves synthetically recovered modalities. These recovered signals may contain various unreliable information (low-fidelity or semantically misaligned reconstructions), and integrating multiple such modalities at once increases the risk of propagating errors or causing representation collapse. The iterative strategy mitigates the risk by conservatively integrating only the most informative recovered modality at each step, ensuring more stable performance (Line 270).
>
> (2) Efficiency. Exhaustively evaluating all combinations of $C$ candidate recovered modalities requires computing MTIR rewards for $2^C$ modality subsets, which is computationally extensive. In contrast, our method only requires computing $C$ MTIR rewards. Prior work (Ma et al., 2019) also shows that such an iterative strategy is both efficient and effective in dynamic modality selection.
>
> Finally, DyMo's outperformance over prior dynamic fusion SOTAs (Fig. 3) and our ablation study results (Tab. 2) show the effectiveness of our modality selection strategy. Though computating MTIR rewards for each individual modality does not explicitly account for potential synergy information from recovered modality combinations, it achieves a favorable accuracy-efficiency balance and provides a reliable and effective solution for our setting.

---

> > ### Author Response · Authors · 2025-11-23
> > **To Reviewer V4dA [Part II]**
> >
> > > W3 & Q3: Impact of the chosen reconstructor to the selection mechanism  ------------------
> >
> > We would like to clarify a misunderstanding: we did **not** use TIP's reconstruction limit to explain the similar performance between CONCAT and DyMo on CAD. Our point is that the similar performance may arise from the comparable amount of task-relevant information provided by TIP's recovered tabular modality across samples (Line 377). Therefore, most dynamic fusion methods (including our DyMo, PDF, DynMM) naturally perform similarly to CONCAT when TIP is used.
> >
> > To directly address the reviewer's concern about a potential reconstruction-quality bottleneck, we additionally evaluate an alternative reconstructor, the iterative multivariate imputer (IMI), a widely-used tabular imputation method that recovers missing values using information from other table columns [Ref1]. Since IMI relies only on table information for reconstruction, it performs worse than TIP (Du et al., 2024).
> >
> > | Model | DVM 0.7 | DVM 0.9 | UKBB 0.7 | UKBB 0.9 |
> > |-------|-----|-----|-----|-----|
> > | CONCAT (IMI) | 81.85 | 67.13 | 73.75 | 69.39 |
> > | DyMo$_e$ (IMI) | 95.97 | 94.28 | 75.96 | 71.09 |
> > | CONCAT (TIP) | 94.74 | 86.08 | 75.91 | 72.04 |
> > | DyMo$_e$ (TIP) | **96.81** | **94.76** | **76.14** | **72.96** |
> >
> > With this weaker reconstructor, CONCAT suffers a large accuracy drop compared to its performance with TIP (e.g., -18.95% accuracy on DVM with 90% missing tabular features). In contrast, DyMo remains stable and achieves the best performance under both reconstructors.
> >
> > These results show that DyMo is **not** constrained to the reconstruction-quality bottleneck. This observation is consistent with our generalizability analysis and recovered modality analysis (Appendix C.2, Tab. S5, Appendix C.6), where DyMo paired with various reconstructors (MoPoE/CMVAE/MMVAE++) consistently outperform prior dynamic/incomplete multimodal SOTAs.
> >
> > We have added the experiments using IMI in Appendix C.2 and Tab. S6.
> >
> > [Ref1] Liu, Jingchen, et al. "On the stationary distribution of iterative imputations." Biometrika 2014.

---

> > > ### Comment · Reviewer_V4dA · 2025-11-26
> > >
> > > Thank you for the response. Most of my concerns have been addressed. I will keep my score.

---

> > > > ### Author Response · Authors · 2025-11-27
> > > > **To Reviewer V4dA [Thank you]**
> > > >
> > > > Thank you for taking the time to review our rebuttal. We sincerely appreciate your follow-up and are glad that our clarifications addressed most of your concerns. Thank you again for your thoughtful assessment and consideration.

---

### Official Review · Reviewer_orCx · 2025-11-03

**Soundness:** 2
**Presentation:** 3
**Contribution:** 3
**Rating:** 6
**Confidence:** 3

**Summary:**

The paper tackles incomplete multimodal classification. It introduces an inference-time dynamic modality selection framework that (i) reconstructs missing modalities with an external recovery model, then (ii) selects only the “reliable” reconstructions to fuse with available modalities. The key idea is a multimodal task-relevant information reward (MTIR) as a proxy for information gain, plus a latent-space metric calibration to guard against low-fidelity reconstructions. Experiments show gains over both recovery-free and recovery-based baselines, especially in severe missingness.

**Strengths:**

1. Clear problem framing (“discarding–imputation dilemma”) and a practical angle: selecting only useful reconstructions instead of always discarding or always imputing.
2. The method is grounded in an information-theoretic heuristic that links mutual information and cross-entropy
3. The method has flexible architecture and straightforward training recipe.
4. Experiments cover five datasets and multiple missing-modality regimes. Reported gains are meaningful in high-missingness settings.

**Weaknesses:**

1. Loose MI–CE bound.
(a) Heuristic bound. The proposed lower bound linking task-relevant information I(Y;Z) to the empirical CE loss is very loose and largely heuristic. The bound involves constants G, yet G can be arbitrarily large since the CE loss is unbounded in practice. As a result, the lower bound can collapse to a small or even meaningless value. Moreover, a reduction in CE loss does not necessarily imply increased mutual information.
(b) Dataset-level rather than per-sample validity.
The high-probability guarantee in the bound applies to the randomness of the training dataset \mathcal{D}, not to individual test samples. It does not provide theoretical support for the per-sample reward computation used during inference without many further assumptions.

2. Dependency on recovery quality.
DyMo’s selection quality depends entirely on the fidelity of recovered modalities. When all reconstructions are poor, the model effectively reverts to observed modalities while still incurring extra inference cost.

3. Task scope.
The method is evaluated for classification. Since many multimodal applications are detection/segmentation/seq-to-seq, a short discussion of what’s needed to extend MTIR beyond CE classification would strengthen the impact.

**Questions:**

1. In Section 3.1, could the “dummy token” embeddings still introduce bias into positional encodings if not learned or masked carefully?

2. In Section 3.2, the computation of class prototypes as arithmetic means assumes locally Euclidean and unimodal feature geometry. This assumption may not hold in practice. Could you provide statistics or visualizations showing that class cluster means are representative?

3. Why model intra-class distances with a truncated normal distribution? Heavy-tailed or multi-modal classes may violate it. Have you explored nonparametric alternatives (e.g., kernel density) to avoid this assumption? Is there empirical evidence that ICS meaningfully quantifies the representativeness of z within its class cluster?

---

> ### Author Response · Authors · 2025-11-23
> **To Reviewer orCx [Part I]**
>
> We sincerely appreciate your positive and constructive feedback, they are very helpful in improving our paper. We have carefully incorporated them in the revised paper. Please see our responses below, which we hope will effectively address your concerns.
>
> > W1: MI-CE bound concern ------------------
>
> (a) Heuristic bound
>
> We emphasize that our MI-CE bound is a lower bound intended for inference-time usage. Both MI and CE loss in the bound are discussed on an IID test dataset with a converged model. Under this setting, this bound is **not** heuristic in practice. Assuming that the CE loss at test time is finite is common and is adopted in prior work (Cao et al., 2024; Zhang et al., 2023; Mori et al., 2018). Our MI-CE lower bound is formulated as follows (Eq. 2 and Eq. S9) is:
>
> $$
> I(Y,Z) \geq H(Y) - \hat{L}_{ce} - G \sqrt{\ln{(1/\delta)}/2|D|}
> $$
>
> , with probability at least $1-\delta$. In this expression, the CE loss is the dominant varying term, since the first entropy term and the third $G$-related term are constant. Importantly, the $G$-related term is scaled by $1/\sqrt{|D|}$ (the test dataset size), meaning it becomes negligible for a reasonably large test set.
>
> To directly address the reviewer’s concern, we show the test CE loss range for DyMo on PolyMNIST with 60% missing modalities (taking $G$ as the maximum CE loss and $\delta=0.1$).
>
> |Dataset| Size | Min | Max (G) | G-related term in the MI-CE lower bound|
> |------|------|------|-----|-----|
> |PolyMNIST| 7,000 | 0 |  4.14 | 0.0053 |
>
> The results show that the test CE loss stays within a well-behaved numerical range. The $G$-related term is extremely small (<0.006), and thus does **not** affect the tightness of the lower bound in practice. Therefore, G does **not** cause the resulting bound to collapse or become meaningless. We have added these statistics in Appendix C.4 and Tab. S7, and clarified Sec 3.2 and Appendix A.1 that the CE loss is evaluated on “tes”’ data.
>
> Finally, we emphasize that our goal is not to provide an exact numerical estimate of MI, but to establish a monotonic and theoretically grounded relationship between CE loss and task-relevant information. This is consistent with the standard information-theoretic practice (e.g., (Tishby et al. 2015), [Ref1], [Ref2], [Ref3]), where tractable surrogates are used to relate MI to observable losses. While decreasing the CE loss does not guarantee a strict MI increase, it does increase our lower bound, supporting CE as a meaningful proxy for information gain. Our visualizations (Fig. 4, Fig. 5, Fig. S4) empirically show that this bound effectively guides DyMo to capture more multimodal task-relevant information.
>
> (b) On dataset-level vs. per-sample validity
>
> We clarify that DyMo is an inference-time method, and that $\mathcal{D}$ in the bound refers to the test dataset, with $|\mathcal{D}|$ denoting its size (Line 200-206). The MI–CE lower bound (Eq. 2) is defined at the dataset level, because mutual information is inherently an expectation over the test-data distribution. We do not claim that this bound directly provides a per-sample high-probability guarantee.
>
> Our use of per-sample loss term optimization for MI objectives follows a widely adopted practice in information-theoretic learning (e.g., (Tishby et al., 2015), [Ref1], [Ref2]), where dataset-level expectations are estimated via per-sample Monte Carlo estimates. This approximation does not introduce additional assumptions beyond those commonly used in MI-based objectives.
>
> Importantly, DyMo does **not** rely on the theoretical bound to assert per-sample guarantees. Instead, the MI–CE bound serves to motivate our inference-time criterion: reducing the per-sample CE loss provides an unbiased estimator of reducing the expected test loss, thereby increasing the MI lower bound. This aligns with heuristics in prior dynamic approaches (Zhang et al., 2023; Cao et al., 2024).
>
> Finally, our extensive empirical results (Fig. 3, Tab. 1, Fig. 4-5) across diverse datasets show that this per-sample criterion is consistently effective in practice, even though the theoretical guarantee applies at the dataset level. This empirically supports the design choice, while keeping the theoretical statements accurate.
>
> [Ref1] Zhang, Yilan, et al. "Prototypical information bottlenecking and disentangling for multimodal cancer survival prediction." ICLR 2024.
>
> [Ref2] Cheng, Pengyu, et al. "CLUB: A contrastive log-ratio upper bound of mutual information." ICML 2020.
>
> [Ref3] Dorent, Reuben, Polina Golland, and William Wells III. "Connecting Jensen-Shannon and Kullback-Leibler Divergences: A New Bound for Representation Learning."NIPS 2025.

---

> > ### Author Response · Authors · 2025-11-23
> > **To Reviewer orCx [Part II]**
> >
> > > W2: Dependency on recovery quality  ------------------
> >
> > We argue that DyMo is robust to varying reconstruction quality. Our results show that DyMo remains stable across different recovery methods, e.g., CMVAE, MVAE++, and MoPoE, despite their noticeable reconstruction performance (Tab. S5 and Fig. S6). This shows that DyMo’s selection quality does **not** entirely depend on fidelity of recovered modalities.
> >
> > In addition, when all reconstructions are poor, DyMo naturally falls back to relying on the observed modalities rather than integrating confusing information, which is precisely the desired behavior. For example, in the 2nd row in Fig. 5 and Fig. S4(b), when all recovered modalities are of low quality, DyMo correctly ignores them and maintains accurate predictions, whereas prior models still integrate those unreliable reconstructions and consequently mispredict.
> >
> > We also clarify that DyMo is **not** designed for settings where reconstructions are intentionally poor; instead, it operates on top of standard reconstruction modules, and aims to selectively integrate only informative recovered modalities, rather than blindly discarding/integrating all reconstructions. This selective integration explains why DyMo consistently outperforms both recovery-based and recovery-free methods (Tab. 1, Fig. 3), as well as the baseline variant that integrates all recovery without selection (Tab. 2).
> >
> > For inference cost, we have shown in Appendix C.3 that DyMo only requires a small number of selection iterations (on PolyMNIST with 80% missing modalities, most samples use only 1-2 steps, and the average number of iterations is 1.38). To further complement this, we now report inference latency on PolyMNIST under 80% random-missing setting:
> >
> > | Model | Inference latency (ms/sample) | Acc (%) | Accuracy gap to DyMo  (%) |
> > |-------|-----|-----|-----|
> > | CONCAT | 0.0035 | 83.69 | -13.72|
> > | PDF | 0.0062 | 82.71 | -14.16 |
> > | M3Care | 0.0168 | 87.92 | -8.89 |
> > | MTL | 0.0169 | 91.14 | -5.67 |
> > | DyMo | 0.0469 | 96.82 | 0 |
> >
> > For samples where all reconstructions are poor, DyMo stops after the 1st iteration and the inference latency is around 0.032 ms/sample. These results show that DyMo achieves a favorable tradeoff: it significantly surpasses prior SOTAs while maintaining an acceptable inference cost.
> >
> >
> > > W3: Task scope  ------------------
> >
> > MTIR is presented in the classification setting because (1) CE-based classification is the common setting for information-theoretic analysis of representations (Tishby et al., 2015; Du et al., 2025; [Ref1]), and (2) it allows a clean derivation of the MI-CE lower bound that DyMo builds on.
> >
> > Extending DyMo to other multimodal tasks is generally feasible, as many tasks (e.g., segmentation/detection/seq-to-seq) also rely on likelihood-based objectives. Since MTIR requires an observable likelihood to approximate $H(Y|Z)$, the framework can naturally generalize by replacing the CE term with the corresponding task-specific likelihood. In specific: (1) segmentation: MTIR can operate on the averaged per-pixel CE loss; (2) detection: MTIR can incorporate both classification and localization likelihoods as the task loss; and (3) sequence-to-sequence modelling: MTIR can aggregate token-level CE losses to guide modality selection.
> >
> > As noted in the conclusion, exploring these extensions is part of our future work (Line 490-491). We now include a brief discussion in Appendix D the modifications required for applying DyMo to other multimodal settings.
> >
> >
> > > Q1: Dummy token  ------------------
> >
> > Dummy tokens do **not** introduce positional bias in our settings. First, our incomplete-modality simulation training strategy randomly simulates missing modalities for each sample in every epoch (Line 295-304). This prevents the model from associating any specific positional pattern with missing modalities.
> >
> > Second, we compute the task loss only from the [CLS] token and apply attention masks that prevent the [CLS] token from attending to dummy tokens (Fig. 2). Therefore, the gradients from the loss do not propagate through positions corresponding to missing modalities.
> >
> > Finally, using dummy tokens together with masking is a common practice in masked-language modelling and multimodal transformer architecture (Ma et al., 2022, Devlin et al., 2019), and our implementation follows the same safe design.

---

> > > ### Author Response · Authors · 2025-11-23
> > > **To Reviewer orCx [Part III]**
> > >
> > > > Q2: Prototype computation  ------------------
> > >
> > > The arithmetic mean is commonly used in prototype-based methods due to its simplicity (Snell et al., 2017; [Ref1]), which is effective when class-conditional representations form compact clusters. In our model, such clustering structure is not only empirically observed but also induced by the design of the latent space. Specifically, all modality combinations are projected into a shared latent representation space, and our auxiliary missing-agnositic contrastive objective (Sec 3.3, Eq. 11) encourages the network to encode modality-complete and modality-missing samples from the same class to lie in a unimodal, consistent feature region. This introduces an inductive bias that aligns with our task assumption: samples of the same class should map to a compact region in the latent space regardless of missing modalities. As a result, class means in this latent space become representative prototypes. Our t-SNE visualizations (Fig. S3 in Appendix C.4) further show that the learned latent space indeed exhibits well-separated and compact class clusters.
> > >
> > >
> > > > Q3: Intra-class similarity score  ------------------
> > >
> > > We model intra-class distances with a truncated normal distribution primarily for its simplicity and empirically effectiveness in our experiment. Prototype-based methods commonly assume that class-conditional features follow a multivariate Gaussian distribution (Appendix A.2, (Snell et al., 2017)). Under this assumption, the distance to the class prototype follows a $\mathcal{X}$-distribution, which reduces to a truncated normal distribution in the one-dimensional embedding. For modelling convenience, we adopt the truncated normal form, which enables efficient computation of the intra-class similarity (ICS) via a closed-form CDF.
> > >
> > > To address potential violations such as heavy-tailed or multimodal classes, DyMo incorporates two design choices: (1) a contrastive training loss that explicitly encourages same-class samples to cluster together, reducing the possibility of heavy tails (Sec. 3.3, Eq. 11); (2) subset-specific prototypes that capture feature variations across different modality subsets when calculating the ICS score (Eq. S15, Appendix A.2). Moreover, we do not directly use the raw ICS score; instead, we rely on the ratio of two scores ($\alpha$) for reward calibration, which further stabilizes the measure (Eq. 9, Line 261-268).
> > >
> > > Our latent-space case study (Fig. 4(b-2)) shows that  ICS meaningfully reflects representativeness within its class cluster. When adding a recovered modality, the updated feature may appear closer to the class prototype, yielding a positive r. However, in the illustrated case, the original feature is actually more representative in the class cluster than the updated feature. ICS correctly reflects this discrepancy, converting the positive r into a negative $r*=-0.77$, thereby preventing the model from accepting an unreliable recovery.  Our ablation studies (Table 2) further show the effectiveness of this intra-class similarity calibration (e.g., +3.19% accuracy on MST and +8.53% on CelebA).
> > >
> > > For nonparametric alternatives (e.g., kernel density estimation), we considered them but found them impractical in our setting. They generally require computation over all training samples (Line 251-252) and would introduce more computational overhead than parametric methods [Ref4], especially since the ICS score requires to be computed for each candidate recovered modality.
> > >
> > > [Ref4] Hastie, Trevor, Robert Tibshirani, and Jerome Friedman. "The elements of statistical learning." (2009).

---

### Author Response · Authors · 2025-11-23
**Reply to all Reviewers**

We thank the reviewers for their valuable and positive feedback. All reviewers had consensus on the clear and well-motivated research problem formulation, as well as the comprehensive experimental evaluation across diverse datasets/domains/tasks/missing scenarios (**R1**: orCx, **R2**: V4dA, **R3**: eaDt, **R4**: LF6X). They further recognized our method as practically meaningful (**R1**, **R2**) and innovative (**R3**), highlighting its flexible (**R1**) architecture and comprehensive technical designs (**R3**). Reviewers also appreciated the theoretical grounding (**R1**,**R2**,**R3**), rich ablation studies (**R2**,**R3**), comprehensive visualizations (**R3**), and strong, consistent performance under severe missing-modality conditions (**R1**,**R2**,**R3**).

We have carefully addressed all constructive feedback and made targeted revisions to improve clarity and rigor of our work. All updates have been incorporated into the revised paper, with changes clearly highlighted in blue for your convenience.

We are confident that these improvements further strengthen our paper, and we look forward to hearing any additional thoughts you may have. Once again, we sincerely thank you for your detailed reviews and valuable insights.

---

> ### Author Response · Authors · 2025-12-02
> **Summary for Area Chairs**
>
> We thank the reviewers and the area chairs for their valuable time and thoughtful feedback.
>
> The initial scores were (6, 6, 6, 4). Following our rebuttal, **Reviewer LF6X (R4)** stated in their post-rebuttal comment that their concerns were resolved and that they would increase their score. They subsequently updated their score in the system on Nov 25 (4 -> 6). **Reviewer V4dA (R2)** also noted in their follow-up comment that most concerns were addressed and confirmed maintaining their positive assessment (Nov 26).
>
> Due to the global score reversion following the early closure of the discussion phase, these updates are not reflected in the current system scores, although the reviewers’ post-rebuttal comments remain visible. We provide this context for your reference.
>
> Thank you for your time and consideration.

---

### Meta-Review · Area_Chair_xR9m · 2025-12-10

**Summary:**

All four reviewers view this as a well motivated and practically relevant paper on incomplete multimodal classification. The key idea is DyMo, an inference time dynamic selection of recovered modalities guided by a mutual information inspired reward and an intra class similarity calibration, on top of a flexible multimodal backbone. Initial scores were 6 (orCx), 6 (V4dA), 6 (eaDt), and 4 (LF6X). After rebuttal, LF6X explicitly stated that their concerns were resolved and that they would raise to 6, and V4dA confirmed they would keep a positive assessment. The main concerns centered on the looseness and scope of the MI CE bound, dependence on recovery quality and recovery diversity, choice of distance metric and greedy selection, computational overhead, and coverage of related and contemporary methods. The rebuttal provided detailed clarifications, new experiments, and additional analysis that largely address these points. After reading the paper and the discussion, I agree with the reviewers that DyMo is a solid and timely contribution and I support acceptance.

**Reviewer Concerns:**

orCx (6)

Addressed: Concerns about the MI CE bound being loose and dataset level, the role of the constant G, and the dependence on recovery quality were addressed with a clearer explanation that the bound is used as a monotonic surrogate at test time, empirical CE ranges and G term magnitude, and empirical evidence that DyMo is robust across several recovery models and even in an extreme controlled simulation where reconstructions are noisy. Questions about dummy tokens, prototype means, and the truncated normal ICS modeling were answered and supported by t SNE visualizations and ablations.

Outstanding: The MI bound remains somewhat loose in a strict theoretical sense and does not give per sample guarantees, but this limitation is clearly acknowledged and is acceptable given the strong empirical support.

V4dA (6)

Addressed: Concerns about the choice of distance metric were addressed by clarifying that Euclidean is aligned with the theory, cosine is included as a robustness check, and both variants consistently outperform baselines with small differences. The sub optimality of greedy selection and the potential for modality synergy were discussed, with arguments about reliability and efficiency and supporting ablations showing that the simple greedy scheme works well in practice. The dependence on a particular reconstructor was addressed by adding experiments with an alternative imputer (IMI) and showing DyMo remains strong even when reconstruction quality degrades.

Outstanding: It is still true that greedy selection cannot capture all possible modality synergies, but this is framed as a conscious efficiency tradeoff rather than a hidden assumption.

eaDt (6)

Addressed: Concerns about recovery diversity, computational overhead, calibration benefits, and limitations were addressed with additional experiments using IMI, a detailed comparison of parameters, training time and latency against dynamic baselines, sensitivity analysis for the temperature, and explicit failure case discussion. Questions on sensitivity to recovery quality, scaling with modalities, theoretical constant G, class imbalance, and generalization beyond classification were all answered with quantitative results and a clear outline of how DyMo could extend to other likelihood based tasks.

Outstanding: DyMo is still somewhat more expensive at inference than very simple static baselines, but the tradeoff between accuracy and latency is clearly documented and seems reasonable.

LF6X (4 -> 6)

Addressed: Concerns about whether cross modal noise truly harms, coverage of newer recovery based methods, use of more recent incomplete multimodal baselines, inclusion of more datasets, and the reliance on pseudo labels were addressed with (i) empirical comparisons that show clear degradation when all recoveries are always used, (ii) discussion of diffusion based recovery work and why it is orthogonal to DyMo, (iii) new experiments with CMVAE and explicit discussion of representative recent baselines, (iv) an explanation of why temporal datasets like CMU MOSI are outside the current scope but similar text dominant regimes are already covered by MST and CelebA, and (v) a clearer argument for how the prototype based mechanism deals with misclassification and when it can still fail.

Outstanding: None that are blocking. Reviewer explicitly stated that their concerns were resolved.

**Reviewer Scores:**

orCx: Initial 6. Given the rebuttal and the additional analysis, I expect orCx would keep their score at 6.

V4dA: Initial 6, follow up comment says most concerns are addressed and they will keep their score. I expect V4dA to stay at 6.

eaDt: Initial 6. The rebuttal directly targets their questions with extra experiments and clarifications, so I expect eaDt to remain at 6.

LF6X: Initial 4, post rebuttal comment explicitly says they will raise their score. Based on that, I treat LF6X as 6 after discussion.

All reviewers therefore converge to a positive score of 6, which aligns with my accept decision.

---

### Decision · Program_Chairs · 2026-01-26

Accept (Poster)